

# Contribution of emissions to concentrations: The TAGGING 1.0 submodel based on the Modular Earth Submodel System (MESSy 2.52)

Volker Grewe[1,2], Eleni Tsati[1], Mariano Mertens[1], Christine Frömming[1], and Patrick Jöckel[1]

[1]Deutsches Zentrum für Luft- und Raumfahrt, Institut für Physik der Atmosphäre, Oberpfaffenhofen, Germany
[2]also at: Delft University of Technology, Aerospace Engineering, Section Aircraft Noise and Climate Effects, Delft, Netherlands

*Correspondence to:* Volker Grewe (volker.grewe@dlr.de)

**Abstract.** Questions such as "What is the contribution of road traffic emissions to climate change?" or "What is the impact of shipping emissions on local air quality?" requires a quantification of the contribution of specific emissions sectors to the concentration of radiatively active species and air quality related species, respectively. Here, we present a diagnostics, implemented in the Modular

Earth-System Model MESSy, which keeps track of the contribution of source categories (mainly emission sectors) to various concentrations. The diagnostics is implemented as a submodel (TAGGING) of EMAC (European Centre for Medium-Range Weather Forecasts - Hamburg (ECHAM)/Modular Earth Submodel System (MESSy) Atmospheric Chemistry). It determines the contributions of 10 different source categories to the concentration of ozone, nitrogen oxides, peroxyacytyl nitrate, car-

bon monoxide, non-methane hydrocarbons, hydroxyl and hydroperoxyl radicals (=tagged tracers). The source categories are mainly emission sectors and some other sources for completeness. As emission sectors, road traffic, shipping, air traffic, anthropogenic non-traffic, biogenic, biomass burning, and lightning are considered. The submodel obtains information on the chemical reaction rates, online emissions such as lightning, and wash-out rates. It then solves differential equations for the

contribution of a source category to each of the seven tracers. This diagnostics does not feed back to any other part of the model. For the first time, it takes into account chemically competing effects: For example the competition between $NO_x$, CO, and NMHCs in the production and destruction of ozone. We show that the results are in-line with results from other tagging schemes and provide plausibility checks for concentrations of trace gases such as OH and $HO_2$, which have not previously

been tagged. The budgets of the tagged tracers, i.e. the contribution from individual source categories (mainly emission sectors) to, e.g., ozone, are only marginally sensitive to changes in model resolu-



tion, though the level of detail increases. A reduction in road traffic emissions by 5% shows that road traffic global tropospheric ozone is reduced by 4% only, because the net ozone productivity increases. This 4% reduction in road traffic tropospheric ozone corresponds to a reduction in total

tropospheric ozone by ≈0.3%, which is compensated by an increase in tropospheric ozone from other sources by 0.1%, resulting in a reduction in total tropospheric ozone of ≈0.2%. This compensating effect compares well previous findings. The computational costs of the TAGGING submodel are low with respect to computing time, but a large number of additional tracers are required. The advantage of the tagging scheme is that in one simulation and at every time step and grid point, infor-

mation is available on the contribution of different emission sectors to the ozone budget, which then can be further used in upcoming studies to calculate the respective radiative forcing simultaneously.

## 1 Introduction

Nitrogen oxides ($NO_x$), carbon monoxide (CO), methane ($CH_4$) and non-methane hydrocarbons (NMHC) are precursors of tropospheric ozone ($O_3$). The assessment of the contribution of indi-

vidual emissions of these precursors on air quality and climate requires a detailed analysis of the chemical conversion, transport and deposition of these species in numerical atmosphere-chemistry simulations. A frequently used method is called 'tagging' (Horowitz and Jacob, 1999; Lelieveld and Dentener, 2000; Meijer et al., 2000; Grewe, 2004; Gromov et al., 2010; Grewe et al., 2012). Technically, this method adds a set of diagnostic tracers for each chemical species considered, i.e. one addi-

tional tracer per source category for each chemical species considered. For example, for the species $NO_x$ a set of tagged tracers $NO_x^{ant}$, $NO_x^{rt}$, $NO_x^{shp}$, $NO_x^{air}$, $NO_x^{bio}$, $NO_x^{bb}$, $NO_x^{lig}$, $NO_x^{ch4}$, $NO_x^{n2o}$, and $NO_x^{str}$ is added, which describes the $NO_x$ concentration from anthropogenic non-traffic (e.g. industry, households), road traffic, ships, air traffic, biogenic, biomass burning, lightning, methane and nitrous oxide decomposition and stratospheric ozone production. The idea is that these tagged tracers

experience the same chemical conversions and loss processes (such as deposition), as the simulated tracer $NO_x$. If all emissions of $NO_x$ are considered and tagged, the sum of all tagged diagnostic $NO_x$ tracers equals the simulated $NO_x$ tracer in this approach. A full partition of the simulated tracer concentration with respect to emission sectors can be achieved. Thus, the contribution of an emission sector, such as industry, road traffic, etc. to a concentration is provided by the tagging method.

A different approach, the perturbation approach (e.g. Hoor et al., 2009; Grewe et al, 2007), where results from two simulations are compared which differ in the strength of an individual emission source, identify the impact of changes in emissions (e.g. by mitigation options) on the atmospheric composition. It is important not to confuse both approaches. For example, the change in ozone due to a 100% reduction in road traffic emissions is smaller by a factor of 5 than the contribution of

the road traffic emissions to ozone (Grewe et al., 2012). Emmons et al. (2012) showed that similar results (factor of 3) are obtained for biomass burning $NO_x$ emissions and the impact on ozone.





Clearly, the non-linearity in the ozone chemistry leads to these large differences. Any reduction in $NO_x$ emission leads mostly to a larger ozone production efficiency. Grewe et al. (2012) showed that in the simulation without road traffic $NO_x$ emissions, the obvious large reduction in ozone

from the reduced road traffic contribution to ozone is compensated by larger contributions from other emission sectors, not because these emissions are changed, but because the ozone production efficiency is increased.

These two different approaches answer two different questions. The perturbation approach quantifies how much a concentration changes if emissions are changed, whereas tagging addresses the

contribution of an emission to the concentration. The combination of both approaches leads to much better insights in the reasons how emission changes lead to concentration changes (Grewe et al., 2012). Note also that the perturbation approach often requires the identical meteorology in either simulation to enhance the signal-to-noise ratio enabling a robust signal. However, this is not feasible in fully coupled chemistry-climate models unless run in a so-called "QCTM-mode", which replaces

instantaneous chemical feedbacks by climatological values (Deckert et al., 2011, see also below).

Most tagging approaches address a straight process chain from the emission of e.g. $NO_x$ to a concentration of e.g. ozone. Grewe et al. (2010), as well as Grewe (2013a) and Tsati (2014) proposed a more general tagging approach, where competing mechanisms in the production of ozone can be taken into account, e.g. both $NO_x$ and carbon compounds (CO, $CH_4$, NMHCs) are precursors of

ozone. This more general tagging approach allows the contribution of road traffic $NO_x$, CO, and NMHC emissions to ozone, for example, to be determined. This generalised method has also been successfully applied to a non-chemical application, namely temperature in an energy balance model (Grewe, 2013b).

Here, we present a submodel (TAGGING) of an Earth-System-Model (EMAC), which applies this

general tagging approach to allow the contribution of $NO_x$, CO and NMHC emissions from a variety of emission sectors to ozone and $HO_x$ chemistry to be quantified. In Section 2 we present the basic equations of the tagging scheme, whereas in Section 3 we present what emissions are addressed and how the tagging method is implemented. In Section 4 we show results of a base simulation and compare them with other modelling studies. Since no measurements are available for contributions

of emissions to ozone concentrations, a direct comparison with observational data is not possible. Instead, we show that the results are in agreement with other studies. Since the tagging of $HO_x$ components is new, we discuss those results in more detail, especially with the focus on aviation and shipping emissions. Finally, we address sensitivities of the methodology (Sec. 5), with respect to the resolution and emission changes, and provide a comparison of the perturbation and tagging method.



## 2 Basics on tagging

The tagging approach, which we adopt here is based on Grewe et al. (2010) and Grewe (2013a).
We first describe the basic mechanism and describe in Sec. 3 how this mechanism is applied in
the submodel TAGGING. Exemplarily, we concentrate on the main reaction for tropospheric ozone
production (rate limiting step; $NO_2$ is photolysed and recombines with $O_2$ to form ozone):

$$NO + HO_2 \longrightarrow NO_2 + OH. \tag{R1}$$

The ozone production rate $P_{R1}$ depends on the abundance of NO and $HO_2$, and the reaction rate
coefficient $k_{R1}$ (Reaction R1). The NO concentration in turn depends on emissions of NO from
different emission sectors (here $N$ in total), such as industry and road traffic with the respective
concentration $NO_x^{ind}$ and $NO_x^{rt}$. Thus, the ozone production rate $P_{R1}$ has to be distributed to the
sectors industry, road traffic, etc. This is achieved by a combinatoric redistribution according to the
concentrations of the tagged species of $NO_x$ and $HO_2$. This means that all possible combinations
between a tagged $NO_x$ species and another tagged $HO_2$ species are evaluated and its probability
calculated consistently with the calculation of the chemical production rate $P_{R1}$. This is just a full
partitioning of the production rate $P_{R1}$:

$$P_{R1} = k_{R1}NO\,HO_2 \tag{1}$$

$$= k_{R1}\sum_{i=1}^{N}NO^i \sum_{j=1}^{N}HO_2^j \tag{2}$$

Here $i$ and $j$ represent a counter for all $N$ source categories; Here, we have chosen $N = 10$ source
categories (see Sec. 3) With this combinatorial approach, Grewe et al. (2010) showed (here as an
example for industry) that the reaction rate $P_{R1}^{ind}$, that is the ozone production due to $NO_x$ from
industry ($NO_x^{ind}$) and due to $HO_2^{ind}$ from industry equals:

$$P_{R1}^{ind} = P_{R1}\frac{1}{2}\left(\frac{NO^{ind}}{NO} + \frac{HO_2^{ind}}{HO_2}\right). \tag{3}$$

Note that this includes the reactions of $NO_x^{ind}$ with all other $HO_2$ molecules and vice versa $HO_2^{ind}$
with other $NO_x$ molecules without any double counting. The relevant differential equation for the
tagged species is then

$$\frac{d}{dt}O_3^{ind} = P^{ind} - D^{ind}, \tag{4}$$

where $P^{ind}$ and $D^{ind}$ are the sum of all relevant production and loss terms. With this approach,
Grewe et al. (2010) showed that the sum of all emissions contributions adds up to the total concen-
tration of the respective species. For example, the ozone field is completely partitioned into emission
sectors contributions, if all emission sectors are included, leading to

$$\sum_{i=1}^{N}O_3^i = O_3. \tag{5}$$





This approach is identical to a different formulation, which describes the right hand side of the differential equation more generally as the relative sensitivity of the individual production and loss terms with respect to the emission sector considered (Grewe, 2013a):

$$P_{R1}^{ind} \quad = \quad P_{R1} \frac{S^{ind^T} \nabla_S P_{R1}}{S^T \nabla_S P_{R1}}, \tag{6}$$

where $S$ is the vector of all chemical compounds, e.g.,

$$S^T \quad = \quad (NO_x, CO, NMHC, O_3, ...)^T, \text{and} \tag{7}$$

$$S^{ind^T} \quad = \quad (NO_x^{ind}, CO^{ind}, NMHC^{ind}, O_3^{ind}, ...)^T, \tag{8}$$

and

$$\nabla_S P_{R1} \quad = \quad \frac{d}{dS} P_{R1} \tag{9}$$

providing two different interpretations of the differential equation (4).

To summarise, this tagging approach fully partitions individual chemical fields into the contribution of individual emission sectors. There is no linearisation required and the approach utilises the identical chemical parameterisation as the underlying chemical scheme, with respect to the probability that a reaction occurs. Note that the new aspect of this tagging approach compared to other tagging approaches (Grewe, 2007; Lelieveld and Dentener, 2000; Emmons et al., 2012) is the competing effect of NO$_x$ and carbon compounds in producing ozone. Since the differential equation for the tagging scheme (4) fully relies on the reaction rates and concentrations, the tagging scheme can be implemented independently from the main chemical solver. However, details on many reaction rates have to be transferred from the chemical solver to the tagging scheme.

## 3 Implementation in EMAC/MECO(n)

The objective of the implementation of this tagging scheme is to be able to monitor online, i.e. at every model's timestep, the contribution of individual emission sectors to ozone and OH, allowing a competition between ozone precursors, linearisation to be avoided, and applicable in decadal simulations. The tagging approach requires to quantify all sources of the species considered. Therefore, in addition to the emission sectors considered, there are additional source categories considered, such as ozone produced by photolysis of oxygen in the stratosphere. In the following the base models are described for which the tagging scheme is developed, an overview on the tagging scheme is given, and the tagging chemistry is described.

### 3.1 Description of MESSy, EMAC and MECO(n)

The TAGGING model described here (see also Tsati, 2014) is written as a submodel of the Modular Earth Submodel System (MESSy), which comprises a standard interface to couple different processes, a simple coding standard and a set of different submodels (Jöckel et al., 2005). The TAG-



**Table 1.** Brief description of the submodels used together with the TAGGING submodel. A complete list can be found in the supplement of Mertens et al. (2016).

| Submodel | Description | Reference |
|---|---|---|
| CLOUD | large scale cloud/rain properties | based on Roeckner et al. (2003) see also Jöckel et al. (2006) |
| CONVECT | convective cloud/rain properties and related transport) | Tost et al. (2006a) |
| DDEP | dry deposition of trace gases | Kerkweg et al. (2006a) |
| JVAL | photolysis rates | Landgraf and Crutzen (1998), see also Jöckel et al. (2006) |
| LNOX | lightning $NO_x$ emissions | Tost et al. (2007b) and Grewe et al. (2001) |
| MECCA | tropospheric and stratospheric gas-phase chemistry | Sander et al. (2011) |
| OFFEMIS | prescribed emissions of trace gases | Kerkweg et al. (2006b) (named OFFLEM therein) |
| ONEMIS | online calculated emissions of trace gases | Kerkweg et al. (2006b) (named ONLEM therein) |
| SCAV | wet deposition and scavenging of trace gases | Tost et al. (2006b) |

GING submodel is implemented in MESSy2 (Jöckel et al., 2010) and consists of two parts, the Sub-Model Interface Layer (SMIL) and the SubModel Core Layer (SMCL). The SMIL part is mainly

important for data management, defining and handling the tracers (using the TRACER submodel described in Jöckel et al. (2008)) and the diagnostic output fields using the CHANNEL submodel (Jöckel et al., 2010). The coupling for the necessary input fields are also handled via the CHANNEL submodel. These input fields comprise, for example, lightning $NO_x$ emissions and chemical production/loss-rates from the chemical solver MECCA (Module Efficiently Calculating the Chem-

istry of the Atmosphere, Sander et al. (2011)).

The TAGGING submodel is implemented in EMAC (ECHAM5/MESSy Atmospheric Chemistry) and MECO(n) (MESSyfied ECHAM and the Consortium for Small-Scale Modelling model COSMO nested n-times). While EMAC uses ECHAM5 as a global circulation model, MECO(n) consists of COSMO/MESSy as a regional-scale model with EMAC as the driving model (Kerkweg and Jöckel,

2012a), which are coupled online. The SMCL of the TAGGING submodel is independent of the base model and consists mainly of the code needed to solve the relevant equations. A detailed description of the TAGGING submodel, including individual subroutines of the SMIL and the SMCL, are provided in the supplement. The model set-up is identical to that of Mertens et al. (2016). A detailed list of applied submodels can be found in the supplement of Mertens et al. (2016) (page 42, therein).

Table 1 describes only those submodels, which are of direct relevance for the TAGGING submodel. An evaluation of the model configurations of EMAC and MECO(n) with respect to the chemical composition of the atmosphere can be found in Jöckel et al. (2016) and Mertens et al. (2016).

### 3.2 TAGGING overview: families, emission sectors and workflow

The objective of the tagging scheme is to determine the contribution of emissions from various

sectors. Here, we discriminate between ten different sources, four anthropogenic: non-traffic an-



**Table 2.** Submodels which provide the source terms (emissions or production terms) for the individual emission sectors (first column) and tagged species (columns 2-4).

| Sector | Tagged species with emissions and other sources | | | |
| --- | --- | --- | --- | --- |
| | $NO_y$ | CO | NMHC | $O_3$ |
| —— Anthropogenic —— | | | | |
| Non-traffic | OFFEMIS | OFFEMIS | OFFEMIS | - |
| Road Traffic | OFFEMIS | OFFEMIS | OFFEMIS | - |
| Ships | OFFEMIS | OFFEMIS | OFFEMIS | - |
| Air Traffic | OFFEMIS | OFFEMIS | OFFEMIS | - |
| —— Natural —— | | | | |
| Lightning | LNOX | - | - | - |
| Biogenic | ON-/OFFEMIS | ON-/OFFEMIS | ON-/OFFEMIS | - |
| $N_2O$ | MECCA | - | - | - |
| $CH_4$ | - | - | MECCA | - |
| Strat-$O_3$ | - | - | - | MECCA |
| —— Mixed —— | | | | |
| Biomass Burning | OFFEMIS | OFFEMIS | OFFEMIS | - |

thropogenic (industry, energy, households), road traffic, ships, and air traffic, five natural sources: lightning, emissions from biogenic sources including soils, decomposition of $N_2O$, decomposition of $CH_4$, stratospheric ozone production by photolysis of $O_2$, and a mixed class: biomass burning (see Table 2).

We use a configuration of the chemical scheme MECCA (Sander et al., 2011), which consists of 72 species. We only tag a reduced set of species, which resemble the main species and families for tropospheric chemistry, in order to limit the required memory. Besides CO, $O_3$, peroxyacytyl nitrate (PAN), $HO_2$, and OH, 2 families are considered: $NO_y$ and NMHC which include all chemically active nitrogen compounds (15) and hydrocarbons (42). All together, the tagging scheme consists of

7 species times 10 emission sectors, thus 70 tagged tracers. For each tracer initialisation, transport (except for OH and $HO_2$), emissions, dry and wet deposition, and chemical conversion has to be deduced from the base model (Figure 1). The tagging scheme utilises the EMAC submodels, e.g. for tracer transport, for emissions computed online during the simulation, and for emissions prescribed by inventories (Table 1; for details see supplement), such as industry, road traffic, etc. (Figure 1,

middle column). It further obtains information on online emissions (lightning, soils), dry and wet deposition, background tracers and reaction rates (left column). This information is processed in tagging core routines (right).

    Here, we concentrate on the TAGGING submodel (Figure 1, right column). For the initialisation of the tagged tracers two options are available. First, the variables can be initialised from files, or





second the tagged tracers can be initialised according to their key characteristics. In this case, the tagged stratospheric ozone is initialised by the ozone field above the tropopause and all other tagged ozone fields are zero above the tropopause and vice versa. Below the tropopause, all but the tagged stratospheric ozone tracer, obtain one ninth of the tropospheric ozone concentration.

At each time step during the simulation, the online emissions (soil emissions) are added to the

respectively tagged tracer (Table 2). The emission rate is obtained by recording the concentration of $NO_x$ before and after the calculation of online emissions. The tagged lighting $NO_y$ tracer obtains the same lightning emissions as the chemical NO tracer, which is provided by the lightning submodel LNOX (Tost et al., 2007b; Grewe et al., 2001). Dry and wet deposition is treated as a bulk process. Changes in the concentration of all relevant chemical species are calculated in a practical and simple

manner, by the difference in the respective concentrations before and after dry and wet deposition is calculated. This tendency of the concentration is provided to the tagging submodel and distributed among the tagged species according to their relative contribution to the total concentration.

### 3.3   TAGGING chemistry

The core of the tagging submodel is the distribution of the chemical tendencies to the tagged tracers

as introduced in Sec. 2. Therefore, the individual production and loss terms have to be determined adequately to calculate concentration changes via Eq. (4). Here, we consider effective ozone production and loss terms according to Crutzen and Schmaizl (1983). This implies that a family is considered for ozone (see supplementary material for more details), which includes all fast exchanges between ozone and other chemical species. The ozone production basically requires splitting up an oxygen

molecule. For the identification of ozone production and loss reactions, we apply the tool *ProdLoss* (see supplementary material for more detailed information), which identifies the effective production and loss reactions for a family in the selected chemical mechanism. This family for effective ozone is hereafter referred to as ozone for simplicity. This results in two ozone production terms, which are applied to any tagged ozone field with the exception of stratospheric ozone. This is reaction (R1) and

the combination of reactions of the type (see supplementary material for more detailed information)

$$NO + RO_2 \longrightarrow NO_2 + RO, \tag{R2}$$

with reaction rate $P_{R2}$. The production and loss terms of these tagged ozone fields are then

$$PO_3^{tag} = \frac{1}{2}P_{R1}\left(\frac{NO_y^{tag}}{NO_y} + \frac{HO_2^{tag}}{HO_2}\right) \tag{10}$$

$$+\frac{1}{2}P_{R2}\left(\frac{NO_y^{tag}}{NO_y} + \frac{NMHC^{tag}}{NMHC}\right)$$



$$
\begin{aligned}
DO_3^{tag} \;=\;& \frac{1}{2}P_{R3}\left(\frac{OH^{tag}}{OH} + \frac{O_3^{tag}}{O_3}\right) \\[4pt]
&+ \frac{1}{2}P_{R4}\left(\frac{HO_2^{tag}}{HO_2} + \frac{O_3^{tag}}{O_3}\right) \\[4pt]
&+ \frac{1}{2}P_{R5}\left(\frac{NO_y^{tag}}{NO_y} + \frac{O_3^{tag}}{O_3}\right) \\[4pt]
&+ \frac{1}{2}P_{R6}\left(\frac{NMHC^{tag}}{NMHC} + \frac{O_3^{tag}}{O_3}\right) \\[4pt]
&+ P_{R7}\frac{O_3^{tag}}{O_3},
\end{aligned}
\tag{11}
$$

with $tag$ denoting one of the ten source tags and with the reaction rates $P_{R3}$, $P_{R4}$, $P_{R5}$, $P_{R6}$, $P_{R7}$ referring to the reactions

$$OH + O_3 \;\longrightarrow\; HO_2 + O_2 \tag{R3}$$

$$HO_2 + O_3 \;\longrightarrow\; OH + 2\,O_2 \tag{R4}$$

$$\text{Effective ozone}\quad\text{loss}\quad\text{via } NO_y \tag{R5}$$

$$RO_2 + O_3 \;\longrightarrow\; RO + 2\,O_2 \tag{R6}$$

$$OH + O_3 \;\longrightarrow\; HO_2 + O_2 \tag{R7}$$

The tagged species $NO_y$, CO, NMHC, and PAN are treated similarly and will be discussed here only briefly, while more detailed information is provided in the supplement. Figure 2 sketches the principal relations between the tagged species. Methane (not tagged) is depleted and the chemical products are then tagged as "NMHC from methane". The species in the NMHC family are eventually transformed into CO and further into $CO_2$. The decomposition of $N_2O$ (not tagged) constitutes a source for "stratospheric $NO_y$". Reactions between $NO_y$ and NMHCs form PAN (not included in $NO_y$). PAN is an important species which can be transported over long distances before it thermally decomposes (Roberts, 2007).

$HO_x$ chemistry (Figure 3 and Table 3) and the calculation of the individual contributions to the concentrations of OH and $HO_2$ is much more complex, hence we discuss it here in more detail. The main source of OH is the reaction of $H_2O$ with $O(^1D)$. The chemical reactions between OH and $HO_2$ involve species such as CO, $CH_4$, $NO_y$, and NMHC. Losses of $HO_x$ are the formation of $H_2O_2$ and $HNO_3$, which are soluble and can be easily rained out.

Since the lifetime of both OH and $HO_2$ is short, we assume steady-state for the contributions. We regard the main $HO_x$ reactions, for which the production and loss rates are calculated in and provided by the MECCA submodel (see also Table 2).





**Table 3.** Reactions and reaction rates used for the calculation of OH and $HO_2$ contributions.

| Reaction | | | OH Prod | OH Loss | $HO_2$ Prod | $HO_2$ Loss |
|---|---|---|---|---|---|---|
| $H_2O + O(^1D)$ | $\longrightarrow$ | 2 OH | $0.5\,P_1^{OH}$ | | | |
| $HO_2 + O_3$ | $\longrightarrow$ | $OH + 2\,O_2$ | $P_2^{OH}$ | | | $L_1^{HO_2}$ |
| $NO + HO_2$ | $\longrightarrow$ | $NO_2 + OH$ | $P_3^{OH}$ | | | $L_2^{HO_2}$ |
| $OH + CO$ | $\xrightarrow{O_2}$ | $HO_2 + CO_2$ | | $L_1^{OH}$ | $P_1^{HO_2}$ | |
| $OH + CH_4$ | $\xrightarrow{O_2}$ | $NMHC + H_2O$ | | $L_2^{OH}$ | | |
| $OH + O_3$ | $\longrightarrow$ | $HO_2 + O_2$ | | $L_3^{OH}$ | $P_2^{HO_2}$ | |
| $OH + NMHC$ | $\xrightarrow{O_2}$ | $NMHC + H_2O$ | | $L_4^{OH}$ | | |
| $OH + HO_2$ | $\longrightarrow$ | $H_2O + O_2$ | | $L_5^{OH}$ | | $L_3^{HO_2}$ |
| $OH + NO_2$ | $\longrightarrow$ | $HNO_3$ | | $L_6^{OH}$ | | |
| $NMHC + NO$ | $\longrightarrow$ | $NMHC + HO_2 + NO_2$ | | | $P_3^{HO_2}$ | |
| $NMHC + HO_2$ | $\longrightarrow$ | $NMHC + O_2$ | | | | $L_4^{HO_2}$ |
| $HO_2 + HO_2$ | $\longrightarrow$ | $H_2O_2 + O_2$ | | | | $L_5^{HO_2}$ |

The steady-state assumption for the contributions to the OH and $HO_2$ concentrations, i.e., $OH^{tag}$ and $HO_2^{tag}$ implies that the individual production terms equal the individual loss terms:

$$P_{OH}^{tag} = L_{OH}^{tag} \tag{12}$$
$$P_{HO_2}^{tag} = L_{HO_2}^{tag}. \tag{13}$$

Again the more complex part of the tagging chemistry is to derive the production and loss terms. Using the reactions in Table 3 and the approach from Grewe et al. (2010), we obtain for the production and loss of $OH^{tag}$:

$$P_{OH}^{tag} = P_1^{OH}\frac{O_3^{tag}}{O_3} + \frac{1}{2}P_2^{OH}\left(\frac{HO_2^{tag}}{HO_2} + \frac{O_3^{tag}}{O_3}\right)$$
$$+ \frac{1}{2}P_3^{OH}\left(\frac{NO_y^{tag}}{NO_y} + \frac{HO_2^{tag}}{HO_2}\right) \tag{14}$$





$$L_{OH}^{tag} = L_1^{OH} \frac{1}{2} \left( \frac{OH^{tag}}{OH} + \frac{CO^{tag}}{CO} \right) \tag{15}$$

$$+ L_2^{OH} \left( \frac{OH^{tag}}{OH} \right)$$

$$+ L_3^{OH} \frac{1}{2} \left( \frac{OH^{tag}}{OH} + \frac{O_3^{tag}}{O_3} \right)$$

$$+ L_4^{OH} \frac{1}{2} \left( \frac{OH^{tag}}{OH} + \frac{NMHC^{tag}}{NMHC} \right)$$

$$+ L_5^{OH} \frac{1}{2} \left( \frac{OH^{tag}}{OH} + \frac{HO_2^{tag}}{HO_2} \right)$$

$$+ L_6^{OH} \frac{1}{2} \left( \frac{OH^{tag}}{OH} + \frac{NO_y^{tag}}{NO_y} \right)$$

This set of equations includes the assumption that exchanges within a family are fast enough to achieve equally distributed tags among family members. For example, concerning $P_1^{OH}$, the contribution of one source to O($^1$D) equals that of $O_3$, i.e. $\frac{O(^1D)^{tag}}{O(^1D)} = \frac{O_3^{tag}}{O_3}$.

Similarly, we derive the individual production and loss terms for HO$_2$:

$$P_{HO_2}^{tag} = P_1^{HO_2} \frac{1}{2} \left( \frac{OH^{tag}}{OH} + \frac{CO^{tag}}{CO} \right) \tag{16}$$

$$+ P_2^{HO_2} \frac{1}{2} \left( \frac{OH^{tag}}{OH} + \frac{O_3^{tag}}{O_3} \right)$$

$$+ P_3^{HO_2} \frac{1}{2} \left( \frac{NMHC^{tag}}{NMHC} + \frac{NO_y^{tag}}{NO_y} \right)$$

$$L_{HO_2}^{tag} = L_1^{HO_2} \frac{1}{2} \left( \frac{HO_2^{tag}}{HO_2} + \frac{O_3^{tag}}{O_3} \right) \tag{17}$$

$$+ L_2^{HO_2} \frac{1}{2} \left( \frac{HO_2^{tag}}{HO_2} + \frac{NO_y^{tag}}{NO_y} \right)$$

$$+ L_3^{HO_2} \frac{1}{2} \left( \frac{HO_2^{tag}}{HO_2} + \frac{OH^{tag}}{OH} \right)$$

$$+ L_4^{HO_2} \frac{1}{2} \left( \frac{HO_2^{tag}}{HO_2} + \frac{NMHC^{tag}}{NMHC} \right)$$

$$+ L_5^{HO_2} \frac{HO_2^{tag}}{HO_2}.$$

Now the Eqs. (12) and (13) can be written as

$$0 = A^{tag} - L^{OH} \frac{OH^{tag}}{OH} + P^{OH} \frac{HO_2^{tag}}{HO_2} \tag{18}$$

$$0 = B^{tag} + P^{HO_2} \frac{OH^{tag}}{OH} - L^{HO_2} \frac{HO_2^{tag}}{HO_2}, \tag{19}$$





with

$$A^{tag} = P_1^{OH}\frac{O_3^{tag}}{O_3} + \frac{1}{2}P_2^{OH}\frac{O_3^{tag}}{O_3} \qquad (20)$$
$$+\frac{1}{2}P_3^{OH}\frac{NO_y^{tag}}{NO_y}$$
$$-\frac{1}{2}L_1^{OH}\frac{CO^{tag}}{CO} - \frac{1}{2}L_3^{OH}\frac{O_3^{tag}}{O_3}$$
$$-\frac{1}{2}L_4^{OH}\frac{NMHC^{tag}}{NMHC} - \frac{1}{2}L_6^{OH}\frac{NO_y^{tag}}{NO_y}$$

$$B^{tag} = \frac{1}{2}P_1^{HO_2}\frac{CO^{tag}}{CO} + \frac{1}{2}P_2^{HO_2}\frac{O_3^{tag}}{O_3} \qquad (21)$$
$$+\frac{1}{2}P_3^{HO_2}\frac{NMHC^{tag}}{NMHC} + \frac{1}{2}P_3^{HO_2}\frac{NO_y^{tag}}{NO_y}$$
$$-\frac{1}{2}L_1^{HO_2}\frac{O_3^{tag}}{O_3} - \frac{1}{2}L_2^{HO_2}\frac{NO_y^{tag}}{NO_y}$$
$$-\frac{1}{2}L_4^{HO_2}\frac{NMHC^{tag}}{NMHC}$$

$$P^{OH} = \frac{1}{2}\left(P_2^{OH} + P_3^{OH} - L_5^{OH}\right) \qquad (22)$$

$$L^{OH} = \frac{1}{2}\big(L_1^{OH} + 2L_2^{OH} + L_3^{OH} \qquad (23)$$
$$+L_4^{OH} + L_5^{OH} + L_6^{OH}\big)$$

$$P^{HO_2} = \frac{1}{2}\left(P_1^{HO_2} + P_2^{HO_2}\right) \qquad (24)$$

$$L^{HO_2} = \frac{1}{2}\big(L_1^{HO_2} + L_2^{HO_2} + L_3^{HO_2} \qquad (25)$$
$$+L_4^{HO_2} + 2L_5^{HO_2}\big)$$

The Eqs. (18) and (19) can easily be solved resulting in

$$OH^{tag} = \frac{A^{tag}L^{HO_2} + B^{tag}P^{OH}}{L^{OH}L^{HO_2} - P^{OH}P^{HO_2}}OH \qquad (26)$$

$$HO_2^{tag} = \frac{A^{tag}P^{HO_2} + B^{tag}L^{OH}}{L^{OH}L^{HO_2} - P^{OH}P^{HO_2}}HO_2. \qquad (27)$$

The quantity $A^{tag}$ ($B^{tag}$) represents the contribution of chemical tracers (tagged and non-tagged, other than OH and $HO_2$) to the net OH production (net $HO_2$ production). The terms $L^{OH}$ and $P^{OH}$ are primarily contributions to OH loss and production rates, which depend on the contribution to OH ($OH^{tag}$) and $HO_2$ ($HO_2^{tag}$), respectively. Only the reaction of OH with $HO_2$ forming water vapour and molecular oxygen constitutes an exception, since the loss of OH is dependent from both OH and $HO_2$ ($L_5^{OH}$). Therefore, it also contributes to $P^{OH}$ (see Eq. (22)), the last term in equation (18), which depends on $HO_2$. Note that in this case it does not lead to a production but destruction of OH.



## 4   Present-day simulation and comparison to other studies

In this Section, we present results of a present day simulation. An actual validation of the tagging method is not feasible, since only the full quantities can be measured, e.g. the ozone concentration, but not the contribution from individual sources. Therefore, we concentrate on a comparison to
earlier studies. In the following sections we present the simulation set-up and give a plausibility check for contributions to the $HO_x$ concentration based on shipping and aviation, focussing on the ozone concentration.

### 4.1   Simulation set-ups

#### 4.1.1   EMAC

To evaluate the TAGGING submodel we conduct two different simulations, one base simulation with all emissions and a second simulation where we reduced all road traffic emissions by five percent. The set-up follows the "Specified Dynamics Reference Simulation" for the Chemistry Climate Model Initiative, and is identical to the RC1SD-base10a set-up described and evaluated by Jöckel et al. (2016), however, extended by the TAGGING module, which we described above.

The simulation is performed with a spectral resolution of T42 and a vertical resolution of 90 levels (up to 0.01 hPa). For the anthropogenic emissions we use the MACCity emissions dataset with a resolution of $0.5°$ described by Granier et al. (2011). The lightning emissions are calculated online using the parameterisation described by Grewe et al. (2001). Emissions of NO from soil and biogenic origin as well as biogenic isoprene ($C_5H_8$) are calculated online by the MESSy submodel ONEMIS
(described as ONLEM in Kerkweg et al. (2006b)). The submodel ONEMIS uses an algorithm based on Yienger and Levy (1995) for NO and Guenther et al. (1995) for isoprene. The dynamic state of the atmosphere is relaxed towards ERA-Interim reanalysis data (Dee et al., 2011) using a weak Newtonian relaxation ("nudging") of the four prognostic variables temperature, divergence, vorticity and the logarithm of surface pressure (Jöckel et al., 2006). Sea-surface temperature and sea-ice
concentration are taken from ERA-Interim as well.

One important difference of our simulation to the RC1SD-base10a set-up is the use of the QCTM-mode of EMAC (Deckert et al., 2011). This QCTM-mode decouples the chemistry and the dynamics by using monthly climatologies (here derived from the RC1SD-base10a simulation) in the radiation code and for the heterogeneous stratospheric reactions. The application of the QCTM mode is im-
portant for overcoming the problem of a low signal to noise ratio in the case of a direct comparison of a base case simulation with one with a small chemical perturbation, which would be present with a fully coupled system. The dynamical and chemical differences between the RC1SD-base10a and our base simulation are shown in the supplement. The simulation covers the period 2004-2010 and is initialized from the RC1SD-base10a simulation. The first year was used as a spin-up period,
resulting in an evaluation period from 2005-2010.



### 4.1.2 MECO(n)

The COSMO/MESSy simulation shown in Sec. 5.1 covers the European domain, including parts of the Eastern Atlantic and North Africa, with a resolution of $0.44°$ ($\approx 50$ km). Simulated is the period from July 2007 until December 2008, with the six months of 2007 used as spin-up phase. The driving EMAC model is applied at a resolution of T42 with 31 horizontal levels and is relaxed towards ERA-Interim reanalysis as well. The same QCTM mode as described above is applied for EMAC and COSMO/MESSy. Both model instances use the anthropogenic MACCity emissions, as well as online calculated soil/biogenic emissions as described above. The simulation differs, however, from the EMAC simulation described above, by using the lightning parameterisation after Price and Rind (1992) to simulate the lightning $NO_x$ emissions on the global scale. In COSMO/MESSy we use the same emissions as on the global scale by regridding the corresponding emissions from EMAC. We have chosen this approach to have emissions as comparable as possible in both model instances. More detailed information about this simulation, including an evaluation of chemical tracer concentrations, is provided by Mertens et al. (2016).

### 4.2 Contributions of emission sectors to $NO_y$, CO, NMHCs, and $O_3$

The six year annual average contributions of the ten emission sectors to the ozone concentration are shown in Figure 4. We compare these results with an earlier model version, which only tags $NO_y$ and ozone (Grewe, 2007, Figure 5b therein), and to earlier similar studies by Lelieveld and Dentener (2000) and Emmons et al. (2012). This comparison aims at verifying that the implementation of the TAGGING mechanism is correct by comparing contribution patterns and magnitudes. We have to keep in mind that the approach is conceptually different from earlier studies and takes into account all ozone precursor emissions and not only $NO_y$. Hence the individual contributions have to be smaller and no agreement can be expected, except for pattern and magnitude. The only direct intercomparison can be performed for the stratospheric ozone mixed into the troposphere, since this process is independent from any precursors. Here Lelieveld and Dentener (2000); Lamarque et al. (2005); Grewe (2007); Emmons et al. (2012) (hereafter denoted as LD00, L05, G07, and E12, respectively) estimated a 5 to 40% contribution from stratospheric ozone to tropospheric ozone in the Southern Hemisphere and mostly systematically lower values of 10% to 25% in the Northern Hemisphere, while tropical values are below 10% (Table 4). Our simulation shows also a minimum in the tropics and lower values in the Southern Hemisphere compared to the Northern Hemisphere. The mixing ratios for January and July are very similar to those of Emmons et al. (2012, not shown).

Ozone formed from lightning $NO_x$ (Figure 4) shows a maximum in the tropics and upper troposphere and larger contributions in the Southern Hemisphere than in the Northern Hemisphere, which is in agreement with G07 and LD00. The maximum contribution from lightning is around 25-30%



and thus lower than G07 (40%) and LD00 (50%), because here we regard the ozone production of all precursors, whereas in G07 and LD00 only $NO_x$ as a precursor is considered (see above).

Agreement between the studies LD00, G07, E12, and our work can also be found with respect to the contribution of anthropogenic emissions to tropospheric ozone. These emissions (here: anthropogenic non-traffic, road traffic, shipping and aviation) predominantly contribute by 30 to 50% in

the Northern Hemisphere. The ozone contribution from biomass burning peaks in the lower tropical troposphere with values of around 10% to 15% which compares well with G07 and LD00 (20%). Around 15% of the tropospheric ozone originates from methane, which reacts with OH and contributes to NMHC compounds and eventually to CO and $CO_2$.

Figure 5 shows the contribution of the individual emission sectors to the tropospheric budgets

of $NO_y$, CO, NMHC, PAN, and $O_3$. Lightning and non-traffic anthropogenic emissions show the largest contributions to $NO_y$. The emitted $NO_y$ from lightning and aviation remains much longer in the atmosphere compared to a surface source such as non-traffic anthropogenic $NO_y$, since lightning and aviation emit mainly in the upper troposphere. Aviation, shipping, and biomass burning have approximately the same contribution.

The different emission sectors have very different emission characteristics. Some are only emitting $NO_y$, such as lightning, or $NO_y$ and NMHCs, such as most anthropogenic sources. This is well reflected in the budgets (Figure 5). Since $NO_y$ is required to form PAN, the decomposition of PAN also produces $NO_y$ and NMHCs with the original tag, e.g. the lightning tag. This is fully consistent with the chosen tagging approach and leads to minor contributions of non-CO and non-NMHC emitting

emission sectors to the CO and NMHC budgets (lightning, stratosphere, aviation). The formation of PAN and hence contributions to PAN (Figure 5, second row) requires both $NO_y$ and NMHCs. None of the ten emission sectors has a large contribution from both. And hence, the contributions of each of the ten sectors to PAN are almost equally distributed around 10%. One exception is methane, which contributes largely to NMHC concentrations but not to $NO_y$. In addition, the NMHCs from

methane are predominantly occurring in areas with low $NO_y$ background, which reduces the impact on PAN. The contribution to tropospheric ozone (Figure 5, second row) reflects the distribution presented in Figure 4, with major contributions from lightning, stratosphere, anthropogenic non-traffic emissions and methane.

### 4.3 Contribution of emission sectors to $HO_x$ concentrations

In this Section, we present the effects of a surface source (shipping) and a higher altitude atmospheric source (aviation) on their contribution to the $HO_x$ concentrations. We have chosen the Mediterranean Sea for shipping, since it includes pristine areas in the middle of the Sea on the one hand, as well as areas which are largely affected by other sources e.g. in southern France (Marseilles) and Italy (harbour areas such as Genoa, Figure 6), on the other hand.





We have identified four areas (A-D) with different chemical characteristics (Table 5, see also Figure 6): Highly polluted areas with high concentrations of $NO_2$ (A and B) and with large (some) impact from shipping in region A (B); a pristine area with some impact from shipping on $NO_x$ and $O_3$ (C), and a pristine area with a larger values of shipping ozone (D).

    Large $NO_x$ concentrations in the background (A and B) impact the chemistry and net production
efficiencies, i.e. the ozone enhancement per $NO_x$ is decreasing with increasing $NO_x$ concentrations (e.g. Dahlmann et al., 2011). The reaction (R1), which transforms $HO_2$ into OH, in principle increases (decreases) the OH ($HO_2$) concentration in the region where large amounts of shipping $NO_x$ is present. However, this reaction only dominates the OH to $HO_2$ ratio if enough ozone is available for the $HO_x$ production. In region A, the very low ozone concentration due to ozone titration by
$NO_x$ limits the availability of OH and the contribution of shipping $NO_x$ to OH is even negative. Region B is less polluted than region A and has lower values of shipping $NO_x$ and therefore reaction (R1) dominates the OH and $HO_2$ contributions from shipping leading to positive contributions to OH and negative to $HO_2$. The tagged shipping ozone is larger in area D compared to A and B (not shown). This leads to a larger contribution to OH via the main production reaction of $H_2O$ with
$O(^1D)$, where the $O(^1D)$ originates from the tagged ozone (see also Table 3). The close coupling of OH with $HO_2$ also enhances the tagged $HO_2$ especially in region D. These processes then lead to a complex picture. It shows negative contributions to OH in region A, mainly due to low ozone concentration limiting the OH availability which is even more pronounced by shipping emissions. The shipping contribution to $HO_2$ in the polluted areas A and B are negative mainly driven by the
reaction (R1). Large positive contributions of shipping to OH and moderate negative contributions to $HO_2$ are found in region C, resulting from a combination of effects from reaction (R1) and the main OH production resulting from tagged shipping ozone. Whereas in region D moderate positive contributions of shipping to OH and large negative contributions to $HO_2$ are found. Overall, the contributions from shipping emissions to the OH and $HO_2$ concentrations show a complex picture,
which results from variations in both the background concentrations and shipping concentrations. The impact of an enhanced horizontal resolution is discussed for the same situation in Sec. 5.

    Figure 7 shows annual mean contributions of aviation $NO_x$ emissions to OH (left) and $HO_2$ (right). The air traffic contribution to OH peaks at around 10-20 fmol/mol at the main flight altitude. At the surface, there are other secondary peaks, basically at the locations of the airports. Lee et al. (2010)
summarised the work of Grewe et al. (2002) and Köhler et al. (2008) in their Figure 10 and showed 4 atmospheric regions, which are affected differently by air traffic. In the first region (RNOy in their paper), which is mainly the air traffic corridor, the reaction (R1) controls the chemical impact from air traffic emissions. This implies that air traffic largely contributes to OH and negatively contributes to $RHO_2$ as shown in Figure 7. The region north of RNOy is called RHO2 and the aviation impact
is largely controlled by the reaction of $O_3$ with $HO_2$ (see Table 3). Hence, this reaction leads to a reduction in $HO_2$ without affecting the OH concentration in a similar manner. The region RO3





is located in the lower troposphere and away from the major flight corridor. Here, a significant contribution from air traffic to ozone is found, but not so much to $NO_y$ (not shown). The region is controlled by an increase in ozone. Hence it leads to a general increase in $HO_x$ via the reaction of
$H_2O$ with $O(^1D)$ (see Table 3). This comparison shows that the OH and $HO_2$ contributions form aviation, calculated here, are consistent with the chemical regimes identified in previous studies.

A more detailed view on this tagging mechanism is feasible by applying it to a Lagrangian framework (Grewe et al., 2014). Within the EU-Project REACT4C (Reducing Emissions from Aviation by Changing Trajectories for the benefit of Climate), the $HO_x$ tagging mechanism was implemented
in the same EMAC model version, including a Lagrangian transport algorithm. Aviation-like pulse emissions of $NO_x$ were released at selected points in the atmosphere, and trajectories with these emissions were tagged so that reactions with the background can be determined in detail. Note that aviation is not emitting CO and NMHCs in our simulation, hence the equations look simplified in Grewe et al. (2014) as the values for $CO^{tag}$ and $NMHC^{tag}$ are zero (see Sec. 3.3). Figure 8 shows the
temporal development of several $NO_x$ related species (top and mid) as well as ozone production and loss terms (bottom) for a pulse emission at 45°W, 50°N and 300 hPa. The $NO_x$ emission induces net production of $O_3^{tag}$ (see Eq. 10 in Grewe et al., 2014), mainly via Reaction (R1) and enhanced $HO_x^{tag}$ as calculated via Eqs. (26) and (27). $NO_x$ reacts with OH and forms $HNO_3$, which eventually leads to washout and a reduction of $NO_y^{tag}$ within a few weeks. When $NO_x^{tag}$ is no longer available for
$O_3$-production, $O_3^{tag}$ is subsequently depleted. We denote the chemical regime, where enough $NO_x$ is available to produce larger amounts of ozone with RegNOx and the following regime as RegO3 (see also Figure8). Regarding the destruction of $CH_4^{tag}$, these two regimes are also characterising the two different depletion pathways. First, as long as sufficient $NO_x^{tag}$ is available, $CH_4^{tag}$ is reduced because of an increase of $OH^{tag}$ via Reaction (R1) ($NO_x$ driven $CH_4$ destruction). Second, when
$NO_x^{tag}$ is removed, $OH^{tag}$ is mainly produced via photolysis of $O_3^{tag}$ and the subsequent Reaction $P_1^{OH}$ ($H_2O + O^1D \rightarrow 2 OH$). The tagged OH and $HO_2$ are far lower in the RegO3 regime compared to the RegNOx regime (Figure 8, mid). And consequently, the gradient in the $O_3$ driven $CH_4$ destruction is not as steep. However, due to the longer time period, it dominates the total amount of methane destruction in this case, which can be seen from a budget analysis for the chemical regimes
RegNOx (blue bars) and RegO3 (red bars) given in Figure 9. Note that the trajectory is transported into polar night around day 5, which leads to a reduction of OH and $HO_2$ and a reduction of the photochemical activity. This example shows a reasonable temporal behaviour of the tagged species and it further shows how combining the tagging methodology and a Lagrangian transport algorithm results in a powerful tool, facilitating a detailed analysis of particular processes.



## 5 Sensitivities

In this Section, we investigate if our tagging scheme responds reasonably to changes in resolution (Sec. 5.1) and emissions (Sec. 5.2). In general, there are no strict verification tests other than checking for plausibility and stability.

### 5.1 Higher resolution: MECO(n)

By applying the MECO(n) system (Kerkweg and Jöckel, 2012a, b; Hofmann et al., 2012; Mertens et al., 2016), we have increased the horizontal resolution over Europe by roughly a factor of 5, from a resolution of roughly 200 km times 300 km in EMAC to 50 km times 50 km in the nested grid. Figure 10 shows the contributions of the individual emission sectors to the tropospheric ozone column as a mean over Europe for the coarse resolution (top) and the finer resolution (bottom). Clearly, the individual contributions are very similar in terms of mean values and the seasonal cycle. The finer resolution simulation shows finer resolved structures in the horizontal (not shown), which however do not largely affect the large-scale budgets.

As an example of the effects of finer resolution, we present OH and $HO_2$ contributions from shipping over the Mediterranean Sea, as discussed in Sec. 4.3 and Figure 6. The OH enhancement along shipping routes is much more visible in the finer resolved case (Figure 6, lower left) compared to the lower resolution (upper left). The structures for the OH and $HO_2$ contributions are again similar: A positive contribution to the OH concentration in the area of shipping emissions (B-D) and a decrease in the contribution to OH and $HO_2$ where background $NO_x$ is largely enhanced (region A).

The comparison of the coarser and finer resolution clearly shows that the tagging scheme is stable in its behaviour. Naturally, the finer resolution enables more detailed and finer resolved chemical changes due to emissions to be quantified, but basic structures are reproduced in either resolution. This implies that, depending on the underlying research question, either model can be used.

### 5.2 Emission changes

We performed an additional global simulation with EMAC where we reduced the road traffic emissions by 5%. The simulation set-up hence follows Hoor et al. (2009). This means that the chemical composition and the ozone productivity is different from the base simulation, which leads to roughly 2-3% reduction in the tagged road traffic ozone (not shown). Generally, a reduction of surface $NO_x$ emissions is increasing the ozone productivity (Emmons et al., 2012; Grewe et al., 2012) and consequently a 5% reduction in emissions is expected to lead to significantly less than a 5% reduction in road traffic ozone, which is consistent with our results. Figure 11 shows the relative change in tropospheric ozone induced by the road traffic emission reduction of 5%. The total ozone change of 0.08% (black bar) is a consequence of the reduction of the contribution of road traffic to the tro-



pospheric total ozone by 0.16% and other compensating effects. In total, this leads to a factor of 2
difference between the total ozone change and the road traffic ozone change. The compensating effects are resulting from larger net ozone production rates for the shipping emission sector and other anthropogenic non-traffic emission sectors. This leads to a larger contribution of the anthropogenic non-traffic (dark blue bar) and the (other than road traffic) traffic emission sector (green bar) to total ozone by 0.04% and 0.06%, respectively. Other non-anthropogenic, i.e. natural emission sectors (red bar) reduce this compensation.

The ratio of 2 between the reduction in total ozone and road traffic indicates that a calculation of the road traffic contribution to tropospheric ozone using the perturbation method, i.e. difference between two simulations with changing emissions, underestimates this contribution by exactly this factor of 2. Other studies have shown slightly larger factors, e.g. a factor of 3 for biomass burning $NO_x$ emissions (Emmons et al., 2012) and a factor of 5 for road traffic $NO_x$ emissions. Here, a smaller factor can be expected, since emissions other than $NO_x$ and their impact on ozone are tagged, which reduces the effects from road traffic emission changes. Further, this factor largely depends on the chemical state of the atmosphere, which differ between the simulations. Hence, a direct intercomparison is not possible, however, the results are plausible.

## 6 Conclusions

We present a submodel for the Earth-System Model EMAC, which diagnoses online the contributions of individual source categories (mainly emission sectors) to the concentrations of various trace species. For the first time, we take into account the competition of nitrogen oxides, carbon monoxide and non-methane hydrocarbons for the production and destruction of ozone. We concentrated on 10 source categories and 7 species and families, which are tagged. As a result, we introduced 70 new tracers. The physical and chemical tendencies for these tracers are obtained from other submodels of EMAC, such as the chemistry (MECCA), scavenging (SCAV), etc. The tagging mechanism is distributing the calculated physical and chemical tendencies into the tagged tracer fields. Therefore, the computing time increase by the TAGGING submodel is small, around 10%.

We performed a present-day simulation and showed that the TAGGING submodel provides contributions of individual emission sectors to the concentration of ozone, which roughly agree with previous estimates. A detailed analysis of the calculated contribution of aviation and shipping to OH and $HO_2$ shows reasonable results in different chemical regimes. Changes in the model's resolution shows a stable performance of the TAGGING submodel. Changes in the strength of road traffic emissions yields a decrease in ozone, which is partly compensated for by an increase in ozone from other source categories, since the ozone production efficiency increases, which is in agreement with earlier findings (Grewe et al., 2012; Emmons et al., 2012).



The advantage of this specific tagging scheme is that (1) the effect of ten source categories on ozone and other trace species can be monitored online in one simulation, (2) the competition be-
tween ozone precursors is included, (3) no linearisation is required, and (4) the scheme is applicable for long-term simulations, e.g. over a century. On the other hand, the disadvantage is that (1) the family concept is not flexible and fixed in this specific way, and consequently (2) any change in the set of chemical species requires an adaptation of the TAGGING scheme, and (3) due to memory limitations, a restriction to the main chemical species and families is required.

To summarise, the TAGGING submodel provides a powerful tool to identify the contribution of individual emission sectors to main atmospheric constituents at every grid point and timestep of the simulation and can be further used to derive, for instance, radiative forcings or contribution to air quality information for individual emission sectors.

## 7   Code availability

The Modular Earth Submodel System (MESSy) is continuously further developed and applied by a consortium of institutions. The usage of MESSy and access to the source code is licensed to all affiliates of institutions which are members of the MESSy Consortium. Institutions can become a member of the MESSy Consortium by signing the MESSy Memorandum of Understanding. More information can be found on the MESSy Consortium Website (http://www.messy-interface.org).

*Acknowledgements.*  This study has been conducted in the framework of the DLR projects VEU and WeCare. We gratefully acknowledge the computational resources provided by the Leibniz Supercomputing Centre (LRZ) in Garching and the German Climate Computing Centre (DKRZ) in Hamburg. We are thankful to Roland Eichinger, DLR, for a detailed internal review.

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





**Table 4.** Comparison of different studies with respect to the contribution (%) of stratospheric ozone to the tropospheric ozone concentration. Numbers are rough estimates, only, as taken from published figures. Note that values for L05 are surface values only and percentage values from E12 are estimated from mixing ratios, however a mean value of 17% is given therein. See text for more explanations. SH and NH abbreviates Southern and Northern Hemisphere, respectively.

| Reference | SH | Tropics | NH |
|---|---|---|---|
| LD00 | 40 | 10 | 25 |
| L05 | 20 | <10 | 10 |
| G07 | 5-10 | 10 | 15 |
| E12 | 20 | <5 | 15 |
| This work | 10-15 | 5-10 | 10-20 |

**Table 5.** Qualitative characterisation of four different regions (A-D) in the Mediterranean Sea. A: Southern France; B: Strait of Gibraltar; C: Central Mediterranean Sea; D: Tunesian Coast. See Figure 6 (top row) for the location of the regions. The signs '++', '+', '○', and '-' indicate a qualitative estimate of the respective characteristics, 'very strong/very large', 'strong/large', 'moderate', 'negative'.

| | A | B | C | D |
|---|---|---|---|---|
| Region has polluted background | ++ | + | ○ | ○ |
| Region is impacted by shipping $NO_x$ | ++ | + | + | ++ |
| Region is impacted by shipping ozone | + | + | + | ++ |
| Shipping emissions are converting $HO_2$ into OH via $NO+HO_2 \longrightarrow OH + NO_2$ | ++ | ++ | + | + |
| Shipping ozone produces OH via $O_3 \longrightarrow O(^1D) \xrightarrow{H_2O} OH$ | + | + | + | ++ |
| Contribution of shipping emissions to OH | - | + | ++ | + |
| Contribution of shipping emissions to $HO_2$ | - | - | ○ | - |



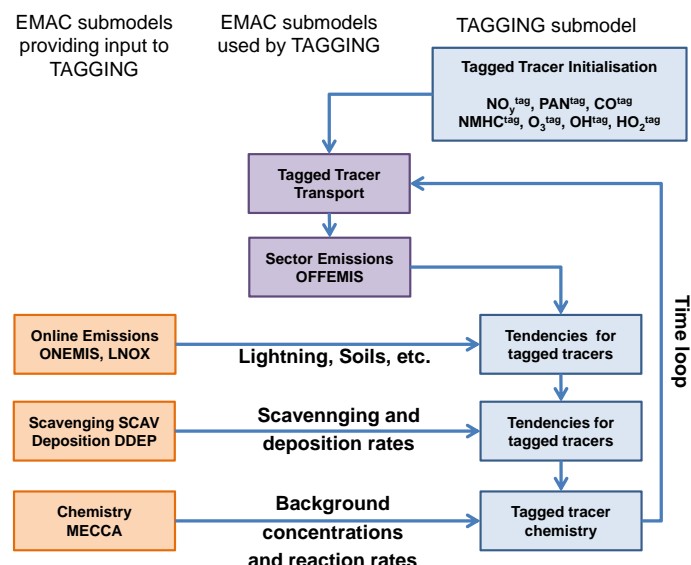

**Figure 1.** Sketch of the tagging algorithm.

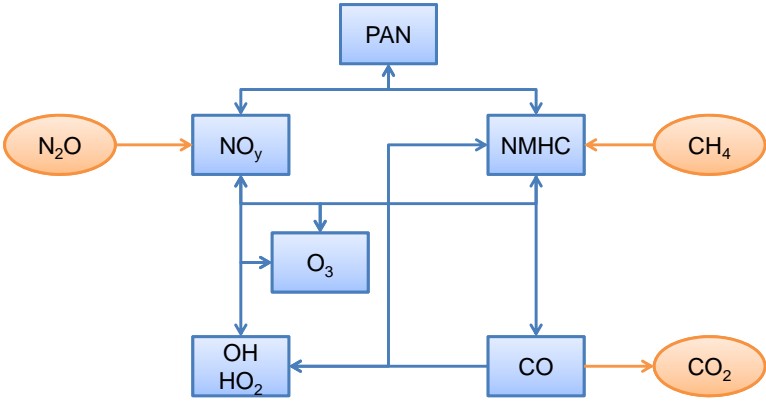

**Figure 2.** Sketch of the chemistry of tagged species (blue) and key relations to other species (orange). Note that stratospheric ozone is not included here. For $HO_x$ chemistry see also Figure 3



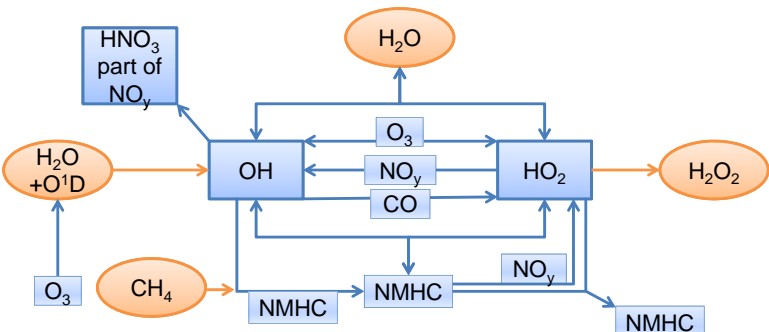

**Figure 3.** Atmospheric $HO_x$ chemistry used in the TAGGING scheme. Blue boxes indicate tagged species and families and orange circles non-tagged species. Arrows indicate reactions.

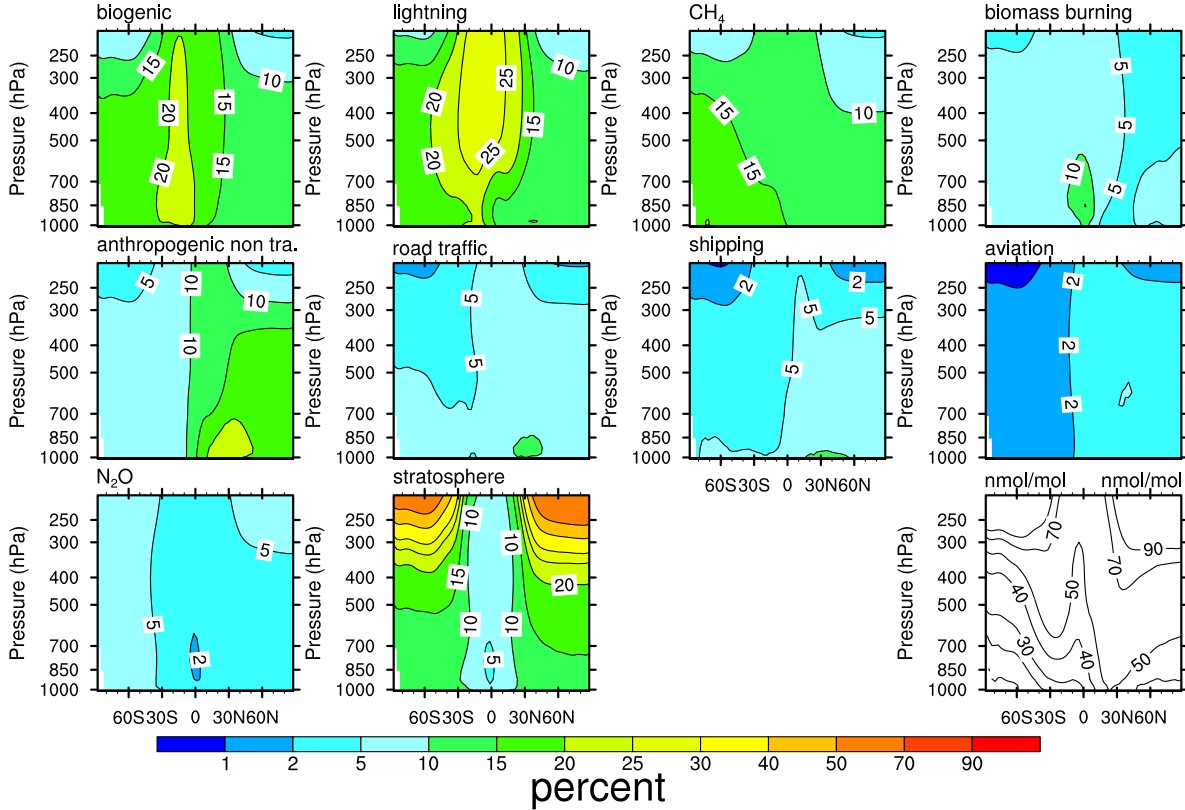

**Figure 4.** Annual mean contributions [%] of 10 emission sectors to the simulated ozone volume mixing ratios. The simulated ozone volume mixing ratio is shown in the lower right panel.





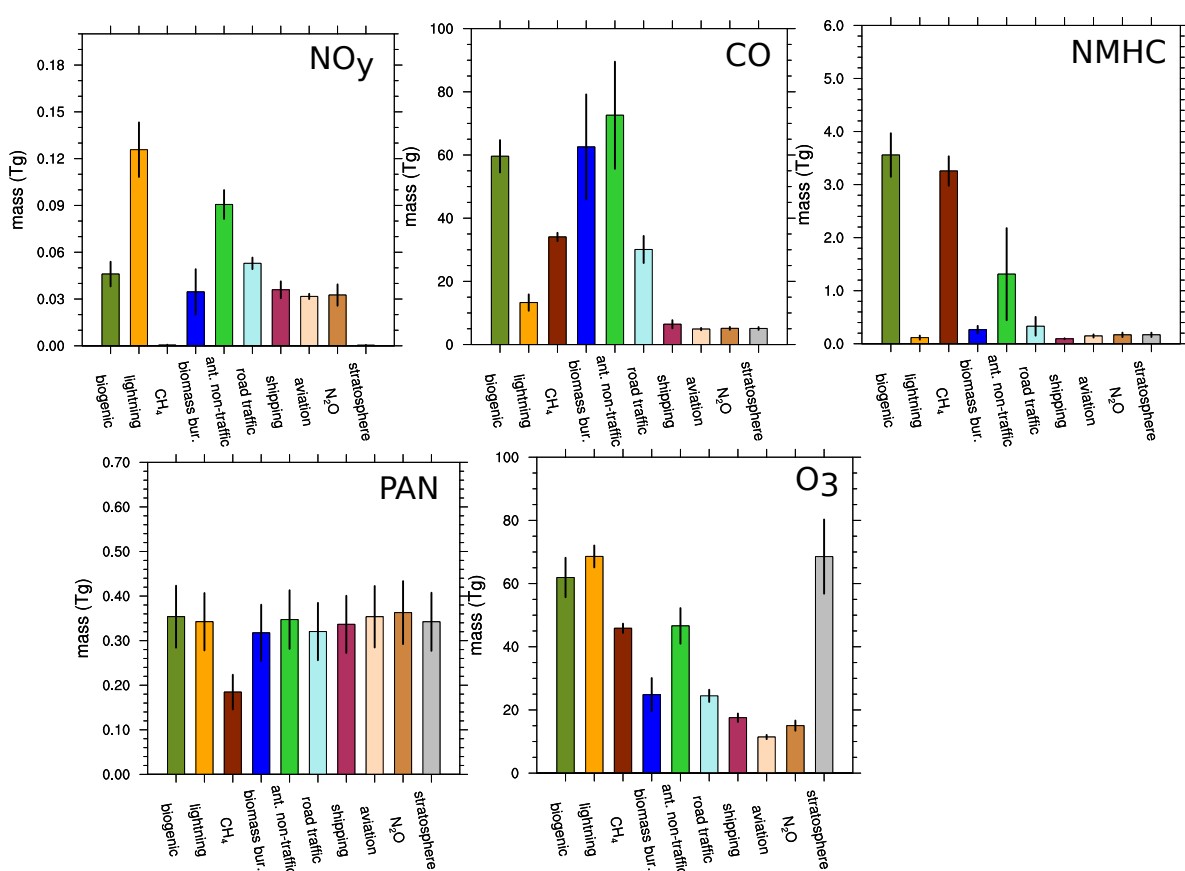

**Figure 5.** Contributions to the annual mean tropospheric budgets [Tg] of 10 emission sectors. Top row: $NO_y$, CO, and NMHCs; Bottom row: PAN and $O_3$. Errorbars indicate the interannual variability.





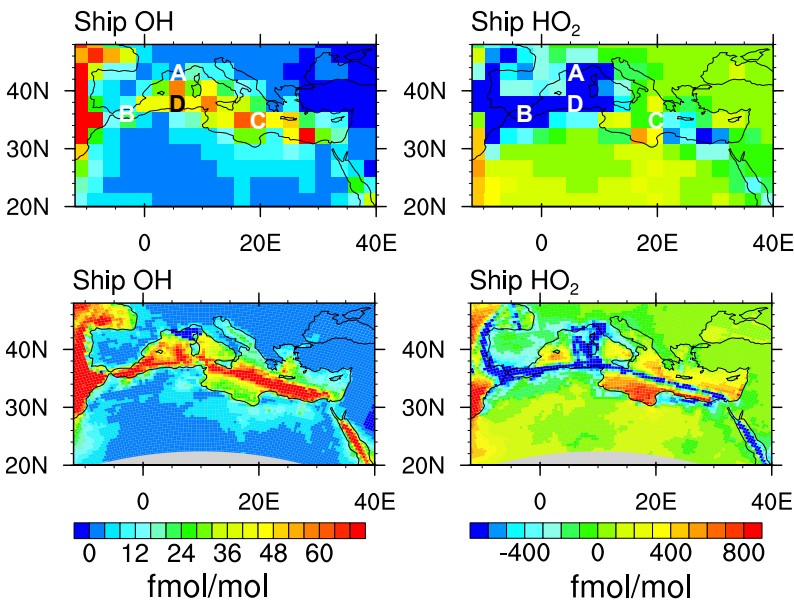

**Figure 6.** Absolute contribution of shipping to the simulated OH (left) and HO$_2$ (right) volume mixing ratios (in fmol/mol) for August 2007. Top row: EMAC; Bottom row: MECO(n). Regions A-D are characterised by different chemical situations. A: Southern France; B: Strait of Gibraltar; C: Central Mediterranean Sea; D: Tunesian Coast; See text for more details.

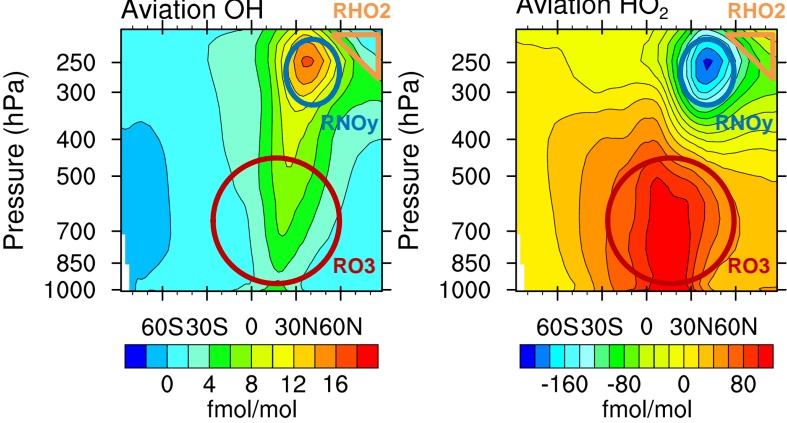

**Figure 7.** Annual mean absolute contribution [fmol/mol] of aviation to the simulated OH (left) and HO$_2$ (right) volume mixing ratios. The regions RO3, RNOy, and RHO2 are characterised by distinct different chemical response to aviation emissions as described by Grewe et al. (2002) (see text for further details).



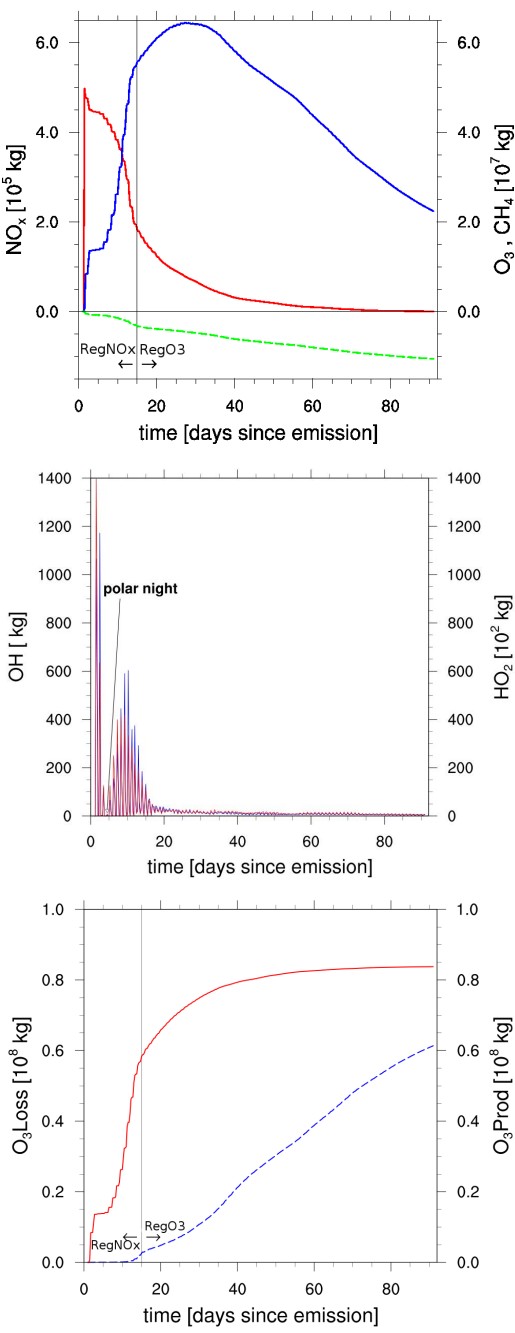

**Figure 8.** Temporal development of $NO_x$ related species (top: $NO_x$ (red), $O_3$ (blue), $CH_4$ (green), mid: OH (blue), $HO_2$ (red)) and production or loss terms (bottom: cumulative $O_3$-loss (blue) and cumulative $O_3$-production (red)) induced by a pulse emission at 45°W, 50°N and 300 hPa on December, 23, 2000. The discrimination between the regimes RegNOx and RegO3 refers to the $NO_x$ dominated (days 1-15 after emission) and the $O_3$ dominated regime (days 16-90 after emission), respectively.





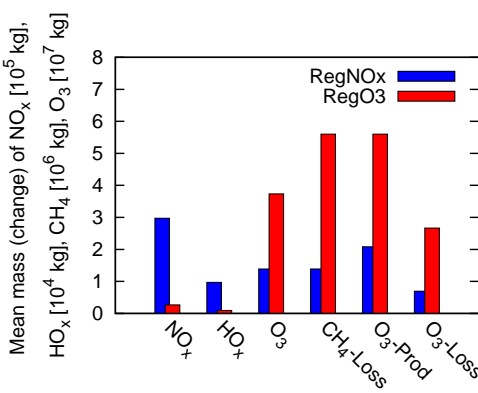

**Figure 9.** Mean contributions to $NO_x$ related species and production or loss terms for the RegNOx regime ($NO_x$ dominated, blue) and RegO3 regime ($O_3$ dominated, red), respectively. Values are given as temporal means over the two time periods.



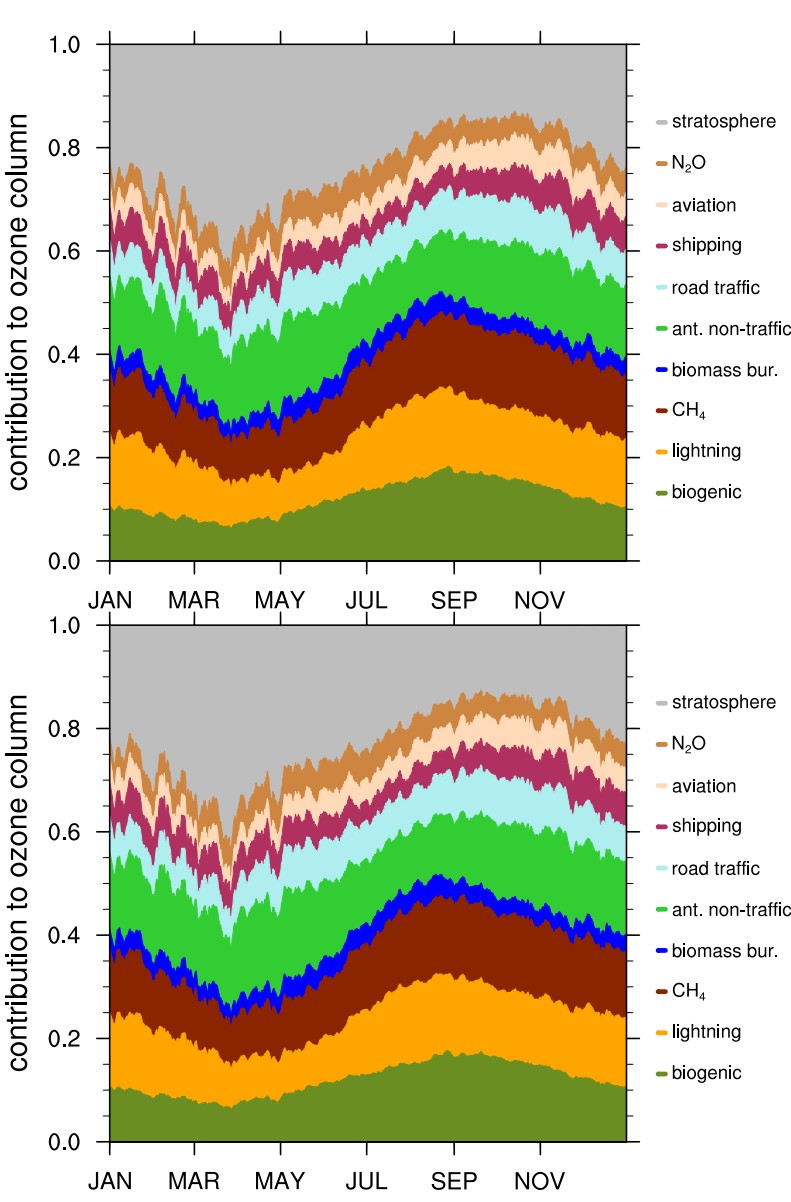

**Figure 10.** Contributions (fraction) of individual emission sectors to the European tropospheric ozone concentration for a coarser resolution simulation with EMAC (top) and a finer resolution with MECO(n) (bottom) for the year 2008.





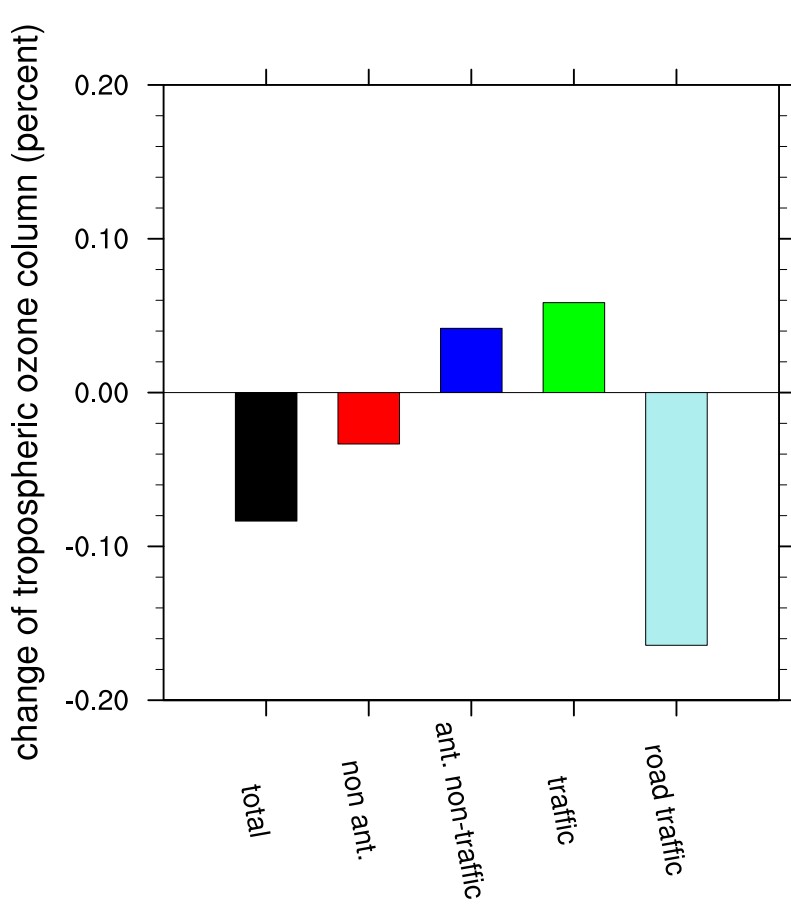

**Figure 11.** Changes in the global tropospheric ozone budget [%] resulting from a 5% reduction in the road traffic emissions.