# Peer review of "Contribution of emissions to concentrations: The TAGGING 1.0 submodel based on the Modular Earth Submodel System (MESSy 2.52)"

_Geoscientific Model Development, 2016_

## Referee Comment (RC1) · Anonymous Referee #1 · 10 Feb 2017

**General comments**

The manuscript by Grewe et al. describes the design and implementation of a novel system for attribution of species concentrations in numerical models to the precursor emissions which produce those concentrations, or "tagging". Some initial results of the system are also presented. Tagging is a useful method for source attribution in numerical models of atmospheric chemistry. There are already a diversity of approaches for such source attribution, including several currently-existing systems which use a tagging approach. This reviewer believes that the community benefits when a large number of diverse approaches to the problem of chemical source attribution exist, and are able to provide results which can be intercompared. For this reason I would ultimately like to see the manuscript published. Before publication however, I believe there appear to be serious issues which the authors should ideally fix, or at least openly acknowledge and thoroughly discuss in their manuscript. There are two problems with the approach as currently described, which lead to the production of unphysical results, which I will describe in more detail below.

One of the most interesting aspects of the TAGGING approach is that it considers both NOx and VOC precursors of ozone simultaneously. Earlier tagging approaches used in scientific applications have tended to focus on just one of these precursors at a time. Emmons et al. (2012), for example attribute all ozone formation to NOx precursors, while Butler et al. (2011) attribute all ozone formation to VOC precursors. The present manuscript attempts to simultaneously attribute ozone formation to both VOC and NOx precursors by using a combinatorial approach that effectively gives equal weight to NOx and VOC precursors. While potentially very interesting, the discussion of this approach ignores the conventional wisdom that tropospheric ozone can be produced under different chemical regimes which are typically referred to as "NOx-limited" or "VOC limited" (see eg. Sillman et al. 1995). Earlier work described by Dunker et al. (2002) uses a technique for attributing ozone production to either NOx or VOCs depending on the chemical regime, which has mostly seen application in regulatory modelling. In order to place their work in more context, the authors should discuss how their approach of equally weighting NOx and VOC precursors of ozone fits with the previous work of Emmons et al (2012), Butler et al. (2011) and Dunker et al. (2002).

An unphysical result stemming from this equal-weight assumption is described by the authors near the end of Section 4.2 and shown for example in Figure 5. The TAGGING submodel attributes a certain proportion of VOC and CO to production from lightning. This is unphysical. Lightning is a source of NOx, not carbon. "CO due to lightning" has no physical meaning, yet is an output of the TAGGING submodel. The authors describe this as "fully consistent with the chosen tagging approach", which while true, omits to mention that this is also unphysical. In a revised version of this manuscript I would

like to see the authors acknowledge this result as being unphysical, and being due to the blending of NOx and VOC precursor tags during the production of tagged PAN (and subsequently produced NOx and VOC products of PAN degradation inheriting this mixture of tags), which is a direct consequence of the equal-weight assumption.

In a future version of the TAGGING scheme, the authors could consider adding 10 additional PAN tracers to their scheme (one per source sector), making it possible to track "PAN from NOx precursors" and "PAN from VOC precursors", and thus reducing these particular unphysical results. Unfortunately PAN is not the only reactive chemical species containing both carbon and nitrogen. For example, most modern chemical mechanisms include one or more alkyl nitrate species. In order to avoid NOx-only tags (such as lightning) being passed on to carbon-containing molecules, duplicate tracers would need to be defined for all kinds of organic nitrates in the model chemical mechanism (or more minimally, just two sets of 10 additional tracers covering an "organic nitrate family"). Their transformations in and out of the VOC and NOy families would also need to be tracked by the TAGGING submodel. Clearly this would add extra complexity to the system, and likely also increase the runtime of the submodel. Low runtime overhead is one of the nice features of the TAGGING submodel as currently described. In their revised manuscript, the authors may wish to discuss this tradeoff between complexity and correctness in their design of the TAGGING scheme.

I believe that a much more serious problem than that described above results from the use of a single chemical "family" to describe all of the VOC species belonging to each tag. This family includes all anthropogenic and biogenic VOCs, their oxidation products, and the oxidation products of methane. The problem with this approach is that not all VOC are created equally. Some VOC are highly reactive in the atmosphere, with very short lifetimes (eg. isoprene), while others have lifetimes orders of magnitude longer (eg. ethane). Different VOC also have different degradation pathways, which can lead to differences in intermediate oxidation products, radical recycling efficiency, and tropospheric ozone production yields between these VOC. I believe that lumping all

of these diverse species together into a single tagged species may result in a significant loss of information about the diverse effects of different classes of VOC, in some cases leading to unphysical results from the TAGGING submodel.

The manuscript does not go into enough detail to describe the way in which this VOC family is treated in the model, in particular how the chemical tendencies obtained from the "real" chemistry are used to modify the concentrations of tagged VOC, and how the effects of "real" reactions involving VOC on radicals, ozone and PAN are distributed to the tagged VOC. Does the TAGGING submodel simply obtain the total VOC tendency in each grid cell from the chemical solver, then apply this tendency to the individually tagged VOC family tracers present in that grid cell? If this is the case, then I see the following problem with this approach: Imagine that a plume of anthropogenically emitted VOC is advected over a forest with large biogenic isoprene emissions. The anthropogenic plume will contain a high fraction of relatively long-lived species such as ethane. With a lifetime of many weeks, such a plume would be capable of being advected over long distances. If a significant amount of isoprene is emitted into this plume, then this will be quickly removed through rapid chemistry, leading to a high negative tendency of the whole VOC family. If this negative tendency is applied equally to each of the tagged VOC species, the result will be that the anthropogenicly tagged VOC is removed at the same rate as the biogenic VOC, leading to an artificially short lifetime for the anthropogenic tag, and an artificially long lifetime for the biogenic tag, thus losing information about the unique properties of each of these VOC sources.

Similarly, the effects of VOC on other species such as radicals and PAN may tend to be smeared, or aliased over the different tags. This effect can actually be seen in Figure 5 of the manuscript, where PAN production has been partially attributed to methane emissions. In both our current understanding of reality, and our current state-of-the-art models of atmospheric chemistry, there is no chemical pathway by which methane emissions can form PAN in the atmosphere. Methane contains one carbon atom. All oxidation products of methane (methyl radical, methyl peroxy radical, formaldehyde,

methyl hydroperoxide, carbon monoxide, carbon dioxide, etc. . . ) also contain one carbon atom. PAN (peroxy acetyl nitrate) contains two carbon atoms. Formation of PAN from methane is unphysical, but the TAGGING submodel nevertheless attributes a proportion of PAN formation to methane. I believe that this unphysical result stems from the use of aggregated family tendencies from the chemical solver being applied equally to each tag.

I would like to see a revised version of the manuscript in which the authors acknowledge that this result (PAN production attributed to methane) is unphysical, explain clearly and in detail how this comes about, offer their thoughts on further unphysical results which may be similarly expected from their approach, and what consequences this has for limiting its usefulness. For example, I believe that the authors should refrain from interpreting the PAN attribution results from the TAGGING system as currently implemented.

In a future version of the TAGGING scheme, the authors could consider adopting approaches used variously by Dunker et al. (2002) to mitigate the problem of different VOC reactivities, and Butler et al. (2011) to ensure that chemical production pathways are respected. Dunker et al. (2002) assign different decay rates to each VOC tag based on the $k_{OH}$ rate constants for each source category, so that (for example) biogenically tagged VOC will decay more quickly than anthrophgenic VOC. Butler et al. (2011) explicitly follow the degradation pathways of each emitted molecule, ensuring that only expected intermediate products are attributed to the original emissions. Both of these approaches would involve an increase in the complexity of the TAGGING submodel. In their revised manuscript, the authors may wish to discuss this tradeoff between complexity and correctness in their design of the TAGGING scheme.

**Specific comments**

Line 15: this diagnostics **package**...

Line 38: Emmons et al. (2012) is already cited below, but should also be listed here as an example of tagging schemes previously used in global models.

Line 40: NOx is technically a chemical family, not a species. Are the authors using the term here as a convenient shorthand for all oxides of nitrogen, or are they describing the implementation of NOx in their model as a chemical family?

Line 94: The reaction following the parenthesised text is not the reaction described in the parenthesised text. This is confusing, please be clearer here about what you mean.

Line 96: Ozone production also depends on RO2.

Line 102: Please provide a forward reference to where tagging of HO2 is described.

Line 184: A table listing the members of the NOy and VOC families would be useful.

Line 217: Please also list the members of this "effective ozone" family.

Table 3: This table appears to be incomplete. Photolysis of formaldehyde should also be an important source of HO2. Is this considered? Are there any other sources left out of this table?

Line 380: Did you mean to write that your simulation shows a lower contribution from stratospheric ozone in the Northern Hemisphere? This would be consistent with the previous work as described in the previous sentence.

[Figure]

**References**

Butler, T., et al.: Multi-day ozone production potential of volatile organic compounds calculated with a tagging approach, Atmos. Env., 45, 4082-4090, 2011.

Dunker, A., et al.: Comparison of source apportionment and source sensitivity of ozone in a three-dimensional air quality model, Environ. Sci. Tech., 36, 2953-2964, 2002.

Emmons, L., et al.: Tagged ozone mechanism for MOZART-4, CAM-chem and other chemical transport models, Geosci. Model Dev., 5, 1531-1542.

Sillman, S.: The use of NOy, H2O2, and HNO3 as indicators for ozone-NOx- hydrocarbon sensitivity in urban locations, J. Geophys. Res., 100, 14175-14188, 1995.

---

## Referee Comment (RC2) · Anonymous Referee #2 · 22 Mar 2017

This new model development by Grewe et alii continues the development of tagging tracers in various ways to attribute environmental degradation to specific emissions (or possibly other actions). The method is reasonable but certainly not a unique or singularly correct approach to derive the environmental damage for a given set of actions. Grewe has made some very specific assumptions about how to partition a key species like tropospheric O3 into a unique sum of causes. The choices made are plausible, but there are readily available other methods (e.g., the 'perturbation method', line 50, which goes back decades before the Grewe references listed here indicate). I tend to concur with the other RC1 review in that a diversity of approaches can always teach us something. Thus, with some chemical model revisions and with a recognition that

tagging does not just give us the "contribution", the model should be published

There are several serious problems with this GMDD paper as written that might be cleared up with major revisions affecting both the (i) chemical modeling and (ii) the authors' choice to describe the tagging method as the true 'correct' method and thus discussing the errors, for example, in the perturbation method. There is an additional (iii) potential problem here in that the extensively expanded TAGGING here may still lack the full chemical coupling across species and regions that has been demonstrated in perturbation experiments with fully coupled models. There are some very interesting results here, but the paper needs to be a bit better balanced and informative.

(i) The chemical model has some clear problems in language or concept. For example, CH4 does NOT make NMHCs. The production of NMHCs (e.g., C2H6 etc) comes from the sources. CH4 is a major source for H2CO in the remote troposphere but that species is not listed here and is not an NMHC. At best it might be a VOC. The NMHC reactions in the table make no sense.

Another mistake appears to be the lack of photolysis of O2 as an important source of O3 in the tropical upper troposphere. This is well established and I can only take it that the old photolysis lookup tables used here have cut off this process in the troposphere? Otherwise the explanation for tropospheric O3 sources does not makes sense. Putting in N2O emissions is interesting, but I do not see where it is then listed as a source of tropospheric NOx ($\sim$0.1 Tg-N/yr)? Moreover, the N2O-CH4-O3 system in the stratosphere is quite complex and controls the net trop influx of O3 and NOy. That system is also established as a coupled perturbation, but is not accurately represented in this tagging model.

Almost all modern scavenging algorithms for species such as H2O2 and HNO3 and others follow the rainout and washout AND re-evaporation of these species in layers below the original uptake. The method of tagging here would seem to be inadequate to deal with this.

(ii) The tagging method as best I understand is built to achieve a zero sum in that a certain level of, say O3 concentration, is partitioned into the different sources so as to sum correctly. This is clearly a personal choice, since I would prefer the linearized tangent approach in which the differential evaluated at the current atmospheric state is used to calculate the change in O3. This of course will not lead to a sum of all the components being zero because the chemistry is non-linear over the range from zero to full industrial emissions. This has a similar issue with CO2 and radiative forcing attribution, since the RF of CO2 goes as the log of its concentration. In this case we have the "which came first?" or "straw that broke the camel's back" problem. Recent increase in CO2 has a lower impact than earlier ones. For many of us the only fair way is to do the perturbation experiment (flat slope at high CO2) and then realize that the sum of all tangent additions if scaled to zero would not be equal to the current state. This is the same problem with the attribution of say biomass burning, and there is no single correct answer. This paper needs to realize and carefully explain the arbitrary choices made.

There is a worrisome statement in the introduction that somehow demonstrates the absurdity of the different approaches. The idea that tagging is the 'correct' answer is just incorrect. It is indeed one of the answers, but not necessarily the best: "For example, the change in ozone due to a 100% reduction in road traffic emissions is smaller by a factor of 5 than the contribution of the road traffic emissions to ozone." The tagging method is obviously defined here to be "the contribution of ", but as a policy maker who wants to know what happens if I reduce road traffic, I would prefer a 100% reduction as the correct answer, or I might choose the 5% reduction times 20. Clearly if I reduce road traffic by 5% or by 100%, the perturbation run, not the tagging run, is the correct value. Does the tagging method do any better than a series of 5% reductions scaled across different emission sources? Moreover, one would not be interested in attributing the background basic atmospheric state (e.g., lightning emissions) in a proportional basis with those of industry, since the background state is not billable for damages as an anthropogenic perturbation is.

The authors continue to use the English word contribution as a code word for their own specific method for dividing up species concentrations. Lines 530-535 argue that the perturbation method "underestimates this contribution" by a factor of 2. I would assert that the "tagging-contribution" is larger than the perturbation by a factor of two and that one of the causes may be the lack of full tagging. The other reason is the apparent need to tag everything including background processes in a similar way to pollution sources. Personally I am not sure that the treatment of the background atmosphere by tagging is done well (e.g., the stratosphere) and thus the partitioning here may be specific to the assumptions made.

Also the authors really need to explain their $\frac{1}{2}$ factor throughout the equations (starting with eqn 3). It is far from obvious since a simple Taylor expansion would not give $\frac{1}{2}$. Please help us out. If it does not apply to small perturbations but only when trying to ensure that the sums are balance, then explain.

(iii) There are clearly identified global chemical coupling patterns that reach across species and regions (strat vs. trop). They are readily identified in models through perturbation simulations. For example, these early chemical feedbacks of tropospheric OH-CH4 and the N2O-NOy-O3 in the stratosphere have been demonstrated to work across many models in various IPCC model comparisons. These are important because they affect the lifetime of a perturbation and hence the attributable damage of emissions. They are most surely in the full MESSY model. From the couplings of this tagging method, I do not believe that these fundamental couplings are present in the tagging model. If you could demonstrate that both of these feedbacks can be derived from the tagging then it would be convincing. Otherwise it shows that tagging really cannot include the dominant chemical feedbacks of the lower atmosphere. This lack of full coupling in the TAGGING model means that one cannot be sure what chemical feedbacks are not included.

Other issues: (iv) The method is described as low cost and non-intrusive, but the only global example given is for T42 resolution (2.8 deg). This is very low resolution for

current global models, yet this is only a GMD paper to establish the development of the model. OK, but can you run at T159 (1.1 deg) for example with all the memory requirements for the tagged tracers to be transported? I had thought that tracer transport was one of the dominant costs of high-res CTMs. Indeed, line 564 seems to indicate that you already have memory limitations at T42.

(v) What was the STE flux of O3 and NOy as a function of latitude and season. This would seem to be very important since the background atmosphere shares the attribution in this scheme. Please denote.

(vi) The idea that the Mediterranean Sea contains "pristine areas" anywhere is at least humorous – thanks.

---

## Author Comment (AC1) · 2 May 2017

**Reply to Anonymous Referee #1:**
We are thankful for the detailed review. While we think that some of the reviewers arguments are based on misinterpretations, they also points at some crucial details of our tagging scheme, which are caused by necessary tradeoff between practicability and accuracy. Assumptions and simplifications are required in such a complex diagnostics. We thought that we have addressed it in our manuscript, but we are happy to discuss in more detail, since it is important to know the limitations. To clarify this we include a new section on the limitations of this approach.

**Reviewer Comment:**
*General comments*
*The manuscript by Grewe et al. describes the design and implementation of a novel system for attribution of species concentrations in numerical models to the precursor emissions which produce those concentrations, or "tagging". Some initial results of the system are also presented. Tagging is a useful method for source attribution in numerical models of atmospheric chemistry. There are already a diversity of approaches for such source attribution, including several currently-existing systems which use a tagging approach. This reviewer believes that the community benefits when a large number of diverse approaches to the problem of chemical source attribution exist, and are able to provide results which can be intercompared. For this reason I would ultimately like to see the manuscript published. Before publication however, I believe there appear to be serious issues which the authors should ideally fix, or at least openly acknowledge and thoroughly discuss in their manuscript. There are two problems with the approach as currently described, which lead to the production of unphysical results, which I will describe in more detail below.*
**Authors' Comment:**
We are happy that the reviewer supports publication of the manuscript, in principle, and we will clarify the two addressed issues (see below).

**Reviewer Comment:**
*One of the most interesting aspects of the TAGGING approach is that it considers both $NO_x$ and VOC precursors of ozone simultaneously. Earlier tagging approaches used in scientific applications have tended to focus on just one of these precursors at a time. Emmons et al. (2012), for example attribute all ozone formation to $NO_x$ precursors, while Butler et al. (2011) attribute all ozone formation to VOC precursors. The present manuscript attempts to simultaneously attribute ozone formation to both VOC and $NO_x$ precursors by using a combinatorial approach that effectively gives equal weight to $NO_x$ and VOC precursors. While potentially very interesting, the discussion of this approach ignores the conventional wisdom that tropospheric ozone can be produced under different chemical regimes which are typically referred to as "NOx-limited" or "VOC limited" (see eg. Sillman et al. 1995).*
**Authors' Comment:**
The reviewer is right that we haven't addressed how the tagging mechanism responses in $NO_x$ and VOC-limited regimes, and instead have focused on other examples. However, the question is indeed intersting and was partly also addressed by Grewe et al. (2010). $NO_x$ and VOC-limited regimes imply that increases of VOC and $NO_x$, respectively, doesn't lead to increases in the ozone production. Still both components

are required for ozone production. While Dunker et al (2002) attributes all ozone produced in a $NO_x$ sensitive regime to the respective $NO_x$ source, we are arguing that either VOCs, CO, or $CH_4$ are necessary to produce ozone and attribute, as the reviewer stated correctly, to both. The factor 0.5, which is a result (not an assumption) of the combinatorial ansatz. Details are, e.g., given by Grewe et al. (2010). As an example we focus on reaction R1 (see manuscript and below). At reaction level, the production, which can be associated to emission category $i$ $(= P_{R1}^i)$ can be split up into 3 parts: $NO^i$ reacts with $HO_2^i$, $NO^i$ reacts with $HO_2^j$ from any other category $(j \neq i)$, and $HO_2^i$ reacts with $NO^j$ from any other category $(j \neq i)$. Note that at reaction level, the reaction of $HO_2^i$ with $NO^j$ for $i \neq j$ is accounted for by 50% to emission category $i$ and $j$:

$$P_{R1}^i = k_{R1}\left(NO^i\ HO_2^i + \sum_{j \neq i}\frac{1}{2}NO^i\ HO_2^j + \sum_{j \neq i}\frac{1}{2}NO^j\ HO_2^i\right) \tag{1}$$

$$= k_{R1}\left(NO^i\ HO_2^i + \frac{1}{2}NO^i(HO_2 - HO_2^i) + \frac{1}{2}HO_2^i(NO - NO^i)\right) \tag{2}$$

$$= \frac{1}{2}k_{R1}\left(\frac{NO^i}{NO} + \frac{HO_2^i}{HO_2}\right). \tag{3}$$

Therefore, the production of ozone via this reaction can be written as

$$P_{R1} = k_{R1}NO\ HO_2 \tag{4}$$

$$= k_{R1}\sum_{i=1}^{N}NO^i\ \sum_{j=1}^{N}HO_2^j \tag{5}$$

$$= \sum_{i=1}^{N}\frac{1}{2}k_{R1}\left(\frac{NO^i}{NO} + \frac{HO_2^i}{HO_2}\right) \tag{6}$$

$$= \sum_{i=1}^{N}P_{R1}^i. \tag{7}$$

At concentration level, the ODE

$$\frac{d}{dt}O_3^i = P^i - D^i, \tag{8}$$

determines the concentration change of $O_3^i$ and implicitely includes effects wrt. $NO_x$ and VOC saturated regimes (see below). The derived factor of 0.5 describes a basic principle and per se is not including explicitly any information on limited regimes. On the other side, the concentrations of ozone and the tagged ozone are revealing these regimes.

As an example, we assume a $NO_y$ concentration of 80 ppbv and VOC of 1.5 ppmv, leading to an ozone steady-state concentration of 100 ppbv. The regime is $NO_x$ limited. Now the tagging scheme attributes 50 ppbv ozone to $NO_y$ and 50 ppbv to VOCs, leading to an attribution of 0.625 ppbv $O_3$ per ppbv $NO_y$ and 33 ppbv $O_3$ per ppbv-VOC. Increasing the VOC emissions leads to 2 ppmv VOC and to the same ozone concentration, since we have a $NO_x$-limited/VOC-sensitive regime. Therefore, the $NO_y$ attribution remains unchanged and the VOC attribution is effectively reduced from 33 to 25 ppbv O3 per ppbv $NO_y$, nicely attributing less ozone per VOC molecule.

Text changes: We now explain this in more detail in Section 2.

**Reviewer Comment:**

*Earlier work described by Dunker et*
*al. (2002) uses a technique for attributing ozone production to either $NO_x$ or VOCs*
*depending on the chemical regime, which has mostly seen application in regulatory*
*modelling. In order to place their work in more context, the authors should discuss how*
*their approach of equally weighting $NO_x$ and VOC precursors of ozone fits with the*
*previous work of Emmons et al (2012), Butler et al. (2011) and Dunker et al. (2002).*

**Authors' Comment:**

The contribution calculations can be divided into two categories, which we refer to as "Perturbation-Method" and "Contribution-Method". Note that there is no consensus in literature about the naming. For the Contribution-Method, tagged species obtain production and loss terms in relation to the tagged source concentration. This method is applied by Lelieveld and Dentener, 2000; Grewe, 2007; Emmons et al., 2012, for the $NO_x$-$O_3$ tagging and by Butler et al., 2012 for the VOC-$O_3$ tagging.

With the "Perturbation-Method" sensitivities are attributed to soures terms. This can be emissions changes, e.g. Hoor et al. (2009). Dunker et al. (2002) used similar techniques as for the "Contribution-Method", but took as production terms the ozone change depending on the chemical regime and by this analyzed ozone sensitivities and thereby achieved a good agreement of their source contribution with calculated sensitivities (see their Figure 4). Hence this method, by definition, is similar to the "Perturbation-Method", since it diagnoses the origin of ozone changes.

Both approaches are answering different research questions and should not be mixed (see also discussion in the Introduction; A detailed analysis between these approaches is given by Grewe et al. (2010) and Grewe et al. (2012).

Text changes: Two text passages are added to the introduction, including the suggested references.

**Reviewer Comment:**

*An unphysical result stemming from this equal-weight assumption is described by the*
*authors near the end of Section 4.2 and shown for example in Figure 5. The TAGGING*
*submodel attributes a certain proportion of VOC and CO to production from lightning.*
*This is unphysical. Lightning is a source of $NO_x$, not carbon. "CO due to lightning" has*
*no physical meaning, yet is an output of the TAGGING submodel. The authors describe*
*this as "fully consistent with the chosen tagging approach", which while true, omits to*
*mention that this is also unphysical. In a revised version of this manuscript I would*
*like to see the authors acknowledge this result as being unphysical, and being due*
*to the blending of $NO_x$ and VOC precursor tags during the production of tagged PAN*
*(and subsequently produced $NO_x$ and VOC products of PAN degradation inheriting this*
*mixture of tags), which is a direct consequence of the equal-weight assumption.*
*In a future version of the TAGGING scheme, the authors could consider adding 10*
*additional PAN tracers to their scheme (one per source sector), making it possible to*
*track "PAN from $NO_x$ precursors" and "PAN from VOC precursors", and thus reducing*
*these particular unphysical results.*

**Authors' Comment:**

The reviewer is right that CO, and equally ozone, OH and $HO_2$, is not emitted by lightning. The tagging scheme shows, however, which species are effectively influenced by individual sources categories.

For example NO$_x$ emitted by traffic may react with hydrocabons from, e.g., biogenic emissions to form PAN, which is then transported over longer distances. After being transported over a long distance it decomposes into NO$_y$ and VOCs. The biogenic emissions contributed to PAN, made a transport over a long distance possible and hence the decomposed NO$_y$ gets a biogenic tag, although it was not initially emitted by the biogenic source, considered. This is meant by "fully consistent with the chosen tagging approach". A different question is whether this is what we want to diagnose. Hence, it is not a question of "physical" or "unphysical", but a question of the objective.

Text changes: We clarify this in Sec. 4.2 and introduce a new Section on limitations.

**Reviewer Comment:**

*Unfortunately PAN is not the only reactive chemical species containing both carbon and nitrogen. For example, most modern chemical mechanisms include one or more alkyl nitrate species. In order to avoid NO$_x$-only tags (such as lightning) being passed on to carbon-containing molecules, duplicate tracers would need to be defined for all kinds of organic nitrates in the model chemical mechanism (or more minimally, just two sets of 10 additional tracers covering an "organic nitrate family"). Their transformations in and out of the VOC and NO$_y$ families would also need to be tracked by the TAGGING submodel. Clearly this would add extra complexity to the system, and likely also increase the runtime of the submodel. Low runtime overhead is one of the nice features of the TAGGING submodel as currently described. In their revised manuscript, the authors may wish to discuss this tradeoff between complexity and correctness in their design of the TAGGING scheme.*

**Authors' Comment:**

Totally agreed. Thanks - that is an important point. The discussion of pros and cons in the conclusion section was meant to include this statement, but unintentionally this got lost.

Text changes: We include a section on limitations.

**Reviewer Comment:**

*I believe that a much more serious problem than that described above results from the use of a single chemical "family" to describe all of the VOC species belonging to each tag. This family includes all anthropogenic and biogenic VOCs, their oxidation products, and the oxidation products of methane. The problem with this approach is that not all VOC are created equally. Some VOC are highly reactive in the atmosphere, with very short lifetimes (eg. isoprene), while others have lifetimes orders of magnitude longer (eg. ethane). Different VOC also have different degradation pathways, which can lead to differences in intermediate oxidation products, radical recycling efficiency, and tropospheric ozone production yields between these VOC. I believe that lumping all of these diverse species together into a single tagged species may result in a significant loss of information about the diverse effects of different classes of VOC, in some cases leading to unphysical results from the TAGGING submodel.*

*The manuscript does not go into enough detail to describe the way in which this VOC family is treated in the model, in particular how the chemical tendencies obtained from the "real" chemistry are used to modify the concentrations of tagged VOC, and how the effects of "real" reactions involving VOC on radicals, ozone and PAN are distributed to the tagged VOC. Does the TAGGING submodel simply obtain the total VOC tendency*

*in each grid cell from the chemical solver, then apply this tendency to the individually tagged VOC family tracers present in that grid cell? If this is the case, then I see the following problem with this approach: Imagine that a plume of anthropogenically emitted VOC is advected over a forest with large biogenic isoprene emissions. The anthropogenic plume will contain a high fraction of relatively long-lived species such as ethane. With a lifetime of many weeks, such a plume would be capable of being advected over long distances. If a significant amount of isoprene is emitted into this plume, then this will be quickly removed through rapid chemistry, leading to a high negative tendency of the whole VOC family. If this negative tendency is applied equally to each of the tagged VOC species, the result will be that the anthropogenicly tagged VOC is removed at the same rate as the biogenic VOC, leading to an artificially short lifetime for the anthropogenic tag, and an artificially long lifetime for the biogenic tag, thus losing information about the unique properties of each of these VOC sources. Similarly, the effects of VOC on other species such as radicals and PAN may tend to be smeared, or aliased over the different tags. This effect can actually be seen in Figure 5 of the manuscript, where PAN production has been partially attributed to methane emissions. In both our current understanding of reality, and our current state-of-theart models of atmospheric chemistry, there is no chemical pathway by which methane emissions can form PAN in the atmosphere. Methane contains one carbon atom. All oxidation products of methane (methyl radical, methyl peroxy radical, formaldehyde, methyl hydroperoxide, carbon monoxide, carbon dioxide, etc. . . ) also contain one carbon atom. PAN (peroxy acetyl nitrate) contains two carbon atoms. Formation of PAN from methane is unphysical, but the TAGGING submodel nevertheless attributes a proportion of PAN formation to methane. I believe that this unphysical result stems from the use of aggregated family tendencies from the chemical solver being applied equally to each tag.*

*I would like to see a revised version of the manuscript in which the authors acknowledge that this result (PAN production attributed to methane) is unphysical, explain clearly and in detail how this comes about, offer their thoughts on further unphysical results which may be similarly expected from their approach, and what consequences this has for limiting its usefulness. For example, I believe that the authors should refrain from interpreting the PAN attribution results from the TAGGING system as currently implemented.*

*In a future version of the TAGGING scheme, the authors could consider adopting approaches used variously by Dunker et al. (2002) to mitigate the problem of different VOC reactivities, and Butler et al. (2011) to ensure that chemical production pathways are respected. Dunker et al. (2002) assign different decay rates to each VOC tag based on the kOH rate constants for each source category, so that (for example) biogenically tagged VOC will decay more quickly than anthrophgenic VOC. Butler et al. (2011) explicitly follow the degradation pathways of each emitted molecule, ensuring that only expected intermediate products are attributed to the original emissions. Both of these approaches would involve an increase in the complexity of the TAGGING submodel. In their revised manuscript, the authors may wish to discuss this tradeoff between complexity and correctness in their design of the TAGGING scheme.*

**Authors' Comment:**

Thanks for pointing this out. We include a Section on limitations and discuss some pros and cons of the

tagging approach in more detail, and discuss future directions to overcome shortcomings.

**Reviewer Comment:**
*Specific comments*
*Line 15: this diagnostics package. . .*
*Line 38: Emmons et al. (2012) is already cited below, but should also be listed here as*
*an example of tagging schemes previously used in global models.*

**Authors' Comment:**
Text adapted accordingly

**Reviewer Comment:**
*Line 40: $NO_x$ is technically a chemical family, not a species. Are the authors using the*
*term here as a convenient shorthand for all oxides of nitrogen, or are they describing*
*the implementation of $NO_x$ in their model as a chemical family?*
**Authors' Comment:**
Here, we have simplified the description. The chemical scheme MECCA is simulating individual species, whereas the diagnostics package TAGGING is summarizing the reaction rates, obtained from MECCA and applied to a family concept. We have rephrased the sentence and clarify that we are talking about a family of reactive nitrogen compounds, which is tagged.

**Reviewer Comment:**
*Line 94: The reaction following the parenthesised text is not the reaction described in*
*the parenthesised text. This is confusing, please be clearer here about what you mean.*
**Authors' Comment:**
Text adapted

**Reviewer Comment:**
*Line 96: Ozone production also depends on RO2.*
**Authors' Comment:**
Correct. Though here we are concentrating exemplarily on one reaction, only, as clearly indicated in this paragraph. The regarded reaction, as written, depends on NO and $HO_2$.

**Reviewer Comment:**
*Line 102: Please provide a forward reference to where tagging of HO2 is described.*
**Authors' Comment:**
included

**Reviewer Comment:**
*Line 184: A table listing the members of the $NO_y$ and VOC families would be useful.*
*Line 217: Please also list the members of this "effective ozone" family.*
**Authors' Comment:**
A table is given in the supplement, which includes the ozone family. We include a reference to the

supplement.

**Reviewer Comment:**
*Table 3: This table appears to be incomplete. Photolysis of formaldehyde should also*
*be an important source of HO2. Is this considered? Are there any other sources left*
*out of this table?*

**Authors' Comment:**
This reaction is included in MECCA, but it is not included in the present version of the TAGGING scheme. Though, most of the OH production (globally) is covered by the chosen set of reactions. However, the reviewer is right that this reaction is indeed important for the OH budget and we have identified the reduced set of reactions as a shortcoming of the OH tagging scheme. Currently, a colleague is preparing a follow-up paper (Rieger et al., in preparation) on the OH tagging scheme with a full set of reactions. Here, 'full' refers to the set of reaction used in the MECCA chemistry scheme. First results are promising and do locally show changes; however the global picture and results presented here for $HO_x$, $NO_y$, and ozone are hardly affected. A comment is added to the new Section on limitations.

**Reviewer Comment:**
*Line 380: Did you mean to write that your simulation shows a lower contribution from*
*stratospheric ozone in the Northern Hemisphere? This would be consistent with the*
*previous work as described in the previous sentence.*

**Authors' Comment:**
Unfortunately, the text is misleading. Emmons et al. (2012) give absolute numbers (mixing ratios). Those are lower on the southern hemisphere than on the northern hemisphere, while the relative contributions in percent are larger on the southern hemisphere than on the northern because of the larger background concentrations on the northern hemisphere. Text is adapted.

**Reviewer Comment:**
*References*
*Butler, T., et al.: Multi-day ozone production potential of volatile organic compounds*
*calculated with a tagging approach, Atmos. Env., 45, 4082-4090, 2011.*
*Dunker, A., et al.: Comparison of source apportionment and source sensitivity of ozone*
*in a three-dimensional air quality model, Environ. Sci. Tech., 36, 2953-2964, 2002.*
*Emmons, L., et al.: Tagged ozone mechanism for MOZART-4, CAM-chem and other*
*chemical transport models, Geosci. Model Dev., 5, 1531-1542.*
*Sillman, S.: The use of $NO_y$, H2O2, and HNO3 as indicators for ozone-$NO_x$- hydrocarbon*
*sensitivity in urban locations, J. Geophys. Res., 100, 14175-14188, 1995.*

**Authors' Comment:**
References included.

---

## Author Comment (AC2) · 2 May 2017

**Reply to Anonymous Referee #2:**
We are thankful for the detailed review. We like to point out two important comments, which can be summarized as a better description of limitations (similar to the reviewer #1) and a better explanation of the factor 1/2 in the final tagging equation. We have included a section on limitations to clarify the reviewer's questions and we included some more equations in the mathematical re-formulation concerning the factor 1/2. For a couple of other comments, however, we have the feeling that the reviewer assumes and even demands that the tagging diagnostics package performs identical to the perturbation approach. This is neither our intention nor do we pretend it. In contrast, we clearly described the difference and stated that a combination of both approaches gives a good insight in atmospheric processes.

**Reviewer Comment:**
*This new model development by Grewe et alii continues the development of tagging*
*tracers in various ways to attribute environmental degradation to specific emissions (or*
*possibly other actions). The method is reasonable but certainly not a unique or singularly*
*correct approach to derive the environmental damage for a given set of actions.*
*Grewe has made some very specific assumptions about how to partition a key species*
*like tropospheric O3 into a unique sum of causes. The choices made are plausible,*
*but there are readily available other methods (e.g., the 'perturbation method', line 50,*
*which goes back decades before the Grewe references listed here indicate). I tend to*
*concur with the other RC1 review in that a diversity of approaches can always teach*
*us something. Thus, with some chemical model revisions and with a recognition that*
*tagging does not just give us the "contribution", the model should be published.*

**Authors' Comment:**
We are happy that the reviewer acknowledge, in principle, the publication of this method. We think that it is important to stress that different methods often address different questions. And hence, we agree that no unique, best, or better technique exists in quantifying e.g., anthropogenic impacts on atmospheric chemistry and climate change. We already state in the introduction
"The combination of both approaches leads to much better insights in the reasons how emission changes lead to concentration changes",
and hence clearly agree and support both reviewers view that the more diagnostics we have, the more we learn. We agree that the "perturbation method" has been used previously, frequently, and by many more authors, which we thought is obvious. The reference to our own work was actually meant to strengthen the point that we use both approaches and that we are not trying to rate one over the other, in general. For different purposes, though, one method might be, and actually is, better suited than the other. On the other hand, our experience is that the different aspects of different diagnostic packages are sometimes not well differentiated. Hence, we tried to define a wording in the beginning of the paper, which clarifies our understanding of the wording "contribution" and "change". There is no unique definition. But to our understanding, our choice is at least meaningful. And it is important to acknowledge that there is a difference, which the reviewers are obviously aware of, but others may not.

**Reviewer Comment:**
*There are several serious problems with this GMDD paper as written that might be*
*cleared up with major revisions affecting both the (i) chemical modeling and (ii) the*
*authors' choice to describe the tagging method as the true 'correct' method and thus*
*discussing the errors, for example, in the perturbation method. There is an additional*

*(iii) potential problem here in that the extensively expanded TAGGING here may still lack the full chemical coupling across species and regions that has been demonstrated in perturbation experiments with fully coupled models. There are some very interesting results here, but the paper needs to be a bit better balanced and informative.*

**Authors' Comment:**

We are happy to provide more information and equally happy to adapt the wording to better balance our enthusiasm on our own work.

**Reviewer Comment:**

*(i) The chemical model has some clear problems in language or concept. For example, CH4 does NOT make NMHCs. The production of NMHCs (e.g., C2H6 etc) comes from the sources. CH4 is a major source for H2CO in the remote troposphere but that species is not listed here and is not an NMHC. At best it might be a VOC. The NMHC reactions in the table make no sense.*

**Authors' Comment:**

We have the impression that there might be a misunderstanding, which we probably have caused by using the word 'tagging' for both, the conceptual approach and the way we implemented it in the current model version. The mathematical concept 'tagging' is a decomposition of the equations in order to derive, what we called, contributions. No further assumptions, linearizations, or other limiting processes are applied. Hence, we do not see any problems with the mathematical concept. On the other hand the implementation requires simplifications. These are now better addressed in a Section by its own. We think that this might have caused some irritations. We think that the way we have implemented the mathematical concept of tagging provides useful information. However, work is still required to deal with shortcomings.

**Reviewer Comment:**

*Another mistake appears to be the lack of photolysis of O2 as an important source of O3 in the tropical upper troposphere. This is well established and I can only take it that the old photolysis lookup tables used here have cut off this process in the troposphere? Otherwise the explanation for tropospheric O3 sources does not makes sense.*

**Authors' Comment:**

Ozone is often divided up into stratospheric and tropospheric ozone, which either means ozone produced in the stratosphere and troposphere, respectively, or ozone present in the stratosphere or troposphere, respectively. Here, we are referring to ozone production terms in a way that we call ozone produced by oxygen photolysis, regardless where it happens, as stratospheric ozone production, since it is a process typical for the stratosphere. In the same way, we name ozone production via other chemical reactions, e.g. $NO+HO_2 \longrightarrow NO_2+OH$, as tropospheric ozone. We focus on the type of production terms, regardless of where it happens. However, we know that the one is a typical stratospheric ozone production and the other a tropospheric ozone production term. (Note that we have one chemical mechanism, from the surface to the middle atmosphere.) We think that this is justified, since the primary goal is to discriminate ozone production from surface emissions from other sources. The tropopause region is actually characterized by both processes. A split into also ozone produced in the tropopshere and stratosphere is feasible, but beyond the scope of this work. We have adapted the text in the introduction to Section 3.

**Reviewer Comment:**

*Putting*
*in N2O emissions is interesting, but I do not see where it is then listed as a source of*
*tropospheric NOx (0.1 Tg-N/yr)? Moreover, the N2O-CH4-O3 system in the stratosphere*
*is quite complex and controls the net trop influx of O3 and NOy. That system*
*is also established as a coupled perturbation, but is not accurately represented in this*
*tagging model.*

**Authors' Comment:**

The mechanism requires a complete set of $NO_x$ emissions and loss terms (see introduction to Section 3). Therefore stratospheric NOx production is included. We do not understand the comment with respect to the coupled system.

**Reviewer Comment:**

*Almost all modern scavenging algorithms for species such as H2O2 and HNO3 and*
*others follow the rainout and washout AND re-evaporation of these species in layers*
*below the original uptake. The method of tagging here would seem to be inadequate*
*to deal with this.*

**Authors' Comment:**

No, the 3D-tendencies are obtained from the scavenging submodel. Negative tendencies are interpreted as losses due to washout, positive tendencies are re-evaporation.

**Reviewer Comment:**

*(ii) The tagging method as best I understand is built to achieve a zero sum in that a*
*certain level of, say O3 concentration, is partitioned into the different sources so as*
*to sum correctly. This is clearly a personal choice, since I would prefer the linearized*
*tangent approach in which the differential evaluated at the current atmospheric state*
*is used to calculate the change in O3.*

**Authors' Comment:**

This actually seems to be the source of a misunderstanding. We are not considering changes in ozone. We are not investigating, how ozone would change if we were changing the strength of any emission source. This question, as discussed in our manuscript in the introduction section, would be best answered by the perturbation approach. Diagnostic packages as that favored by the reviewer or the tagging, we are proposing are adding additional information so that the physical and chemical reasons for these ozone changes can be understood. Here, we are investigating one simulation with one specific chemical regime and attributing emissions to ozone concentrations. In our view that is best described by the wording 'contribution of an emission sector to the atmospheric concentration of ozone' and it is different from 'contribution of changes in emission sectors to change atmospheric concentrations of ozone', to which, at least as far as we understand the comments, the reviewer is referring to.

**Reviewer Comment:**

*This of course will not lead to a sum of all the*
*components being zero because the chemistry is non-linear over the range from zero*
*to full industrial emissions. This has a similar issue with CO2 and radiative forcing*

*attribution, since the RF of CO2 goes as the log of its concentration. In this case we
have the "which came first?" or "straw that broke the camel's back" problem. Recent
increase in CO2 has a lower impact than earlier ones. For many of us the only fair way
is to do the perturbation experiment (flat slope at high CO2) and then realize that the
sum of all tangent additions if scaled to zero would not be equal to the current state.
This is the same problem with the attribution of say biomass burning, and there is no
single correct answer. This paper needs to realize and carefully explain the arbitrary
choices made.*

**Authors' Comment:**

We agree that the fundamental problem is the "straw that broke the camel's back". Please note that
there is a difference between ozone and carbon dioxide, which, in our opinion, questions the approach
suggested by the reviewer. It is well established that for a certain NOy concentration any increase in
NOy reduces the net-ozone production (Ehalt and Rohrer, 1994 and many others). Hence an increase
in any NOx emission source potentially leads to a lower ozone concentration. And this is then the case
for all emission sectors. Defining the contributions on the basis of this approach hence leads even to an
overall negative contribution of emissions to ozone. So what process actually produces ozone to close the
budget? Note that in this case, we are considering only the "last straw" as relevant ozone prodcution
terms. In our approach, we argue that air chemistry is not making any difference from which source a
NO molecule has been emitted. A NO molecule from industry or from traffic emissions has the same
likelihood to react with HO2. The reaction kinetics are the same.

**Reviewer Comment:**

*There is a worrisome statement in the introduction that somehow demonstrates the absurdity
of the different approaches. The idea that tagging is the 'correct' answer is just
incorrect. It is indeed one of the answers, but not necessarily the best: "For example,
the change in ozone due to a 100% reduction in road traffic emissions is smaller by
a factor of 5 than the contribution of the road traffic emissions to ozone." The tagging
method is obviously defined here to be "the contribution of ", but as a policy maker who
wants to know what happens if I reduce road traffic, I would prefer a 100% reduction
as the correct answer, or I might choose the 5% reduction times 20. Clearly if I reduce
road traffic by 5% or by 100%, the perturbation run, not the tagging run, is the correct
value. Does the tagging method do any better than a series of 5% reductions scaled
across different emission sources? Moreover, one would not be interested in attributing
the background basic atmospheric state (e.g., lightning emissions) in a proportional
basis with those of industry, since the background state is not billable for damages as
an anthropogenic perturbation is.*

**Authors' Comment:**

We clearly stated that the perturbation method is the adequate method to answer the question how
changes in road traffic emissions affect ozone or any other atmospheric species (see Introduction). So
we still are puzzled how the reviewer got another impression. We are interested in understanding atmo-
spheric chemistry. We are interested in how much lightning is contributing to the ozone concentration.
Not everything is about "billing for damages". Agreed, that is important. But we also think that under-
standing the simulated changes is equally important and our diagnostics package is contributing to this

understanding.

**Reviewer Comment:**

*The authors continue to use the English word contribution as a code word for their own
specific method for dividing up species concentrations. Lines 530-535 argue that the
perturbation method "underestimates this contribution" by a factor of 2. I would assert
that the "tagging-contribution" is larger than the perturbation by a factor of two and that
one of the causes may be the lack of full tagging. The other reason is the apparent
need to tag everything including background processes in a similar way to pollution
sources. Personally I am not sure that the treatment of the background atmosphere
by tagging is done well (e.g., the stratosphere) and thus the partitioning here may be
specific to the assumptions made.*

*Also the authors really need to explain their 1/2 factor throughout the equations (starting
with eqn 3). It is far from obvious since a simple Taylor expansion would not give 1/2.
Please help us out. If it does not apply to small perturbations but only when trying to
ensure that the sums are balance, then explain.*

**Authors' Comment:**

Again and similar to our previous comment, the reviewer implicitly assumes that we are investigating
changes or perturbations, because she or he is mentioning the Taylor-approximation, which is based
on estimating effects of a small perturbation. Since we have foreseen such a discussion, we already
have included a whole Section on tagging basics, where we clearly state that the tagging principle does
not include any approximations or linearizations. An additional explanation on the factor 1/2 is given
above. At reaction level, looking at reaction kinetics, rather than concentration level, where we consider
solutions of a ODE, the reaction rate for the above considerer reaction is P=k [NO] [HO$_2$]. The chemistry
mechanism is not discriminating between NO from different sources. The reaction rate P is not adapted
whether NO from road traffic or from lightning is reacting with HO$_2$. The molecules NO and HO2 are
both equally important for this reaction. See also answer to reviewer 1. We have included a discussion
on this point in Section 2.

**Reviewer Comment:**

*(iii) There are clearly identified global chemical coupling patterns that reach across
species and regions (strat vs. trop). They are readily identified in models through perturbation
simulations. For example, these early chemical feedbacks of tropospheric
OH-CH4 and the N2O-NOy-O3 in the stratosphere have been demonstrated to work
across many models in various IPCC model comparisons. These are important because
they affect the lifetime of a perturbation and hence the attributable damage of
emissions. They are most surely in the full MESSY model. From the couplings of this
tagging method, I do not believe that these fundamental couplings are present in the
tagging model. If you could demonstrate that both of these feedbacks can be derived
from the tagging then it would be convincing. Otherwise it shows that tagging really
cannot include the dominant chemical feedbacks of the lower atmosphere. This lack
of full coupling in the TAGGING model means that one cannot be sure what chemical
feedbacks are not included.*

**Authors' Comment:**

The tagging method, as a diagnostic package, is controlled by the MECCA chemistry, which includes

feedback processes. And can be analyzed, as suggested, by perturbations of the system. In the paper we have described the feedback of NOx emission changes to the ozone production efficiency: A decrease in, e.g., road traffic emission leads to lower NOx concentrations. This may lead to larger net-ozone production rates. Hence this is a negative feedback. The lower NOx emissions (with unchanged) net-ozone production rates leads to lower ozone concentrations. However, since the net-ozone production rates increase, the ozone reduction is reduced. This is a negative feedback process, which can be deduced only in the comparison of two simulations. Another interpretation is that the unchanged, e.g., lightning emissions, also experience the enhanced net-ozone production rates decrease, and hence the ozone produced by lightning NO is larger than in the previous simulation. During the simulation and more importantly also in reality, a NO molecule has no remembrance of its source of emission. Hence, no difference is made between the reaction of a NO molecule with HO2, whether it originates from lightning or road traffic. Feedbacks are always related to changes with respect to a base situation, whereas tagging is analysing a base situation by itself. So we think that asking for feedbacks to be included in the tagging is simply trying to achieve the same results from the tagging method as a perturbation method is providing. However, this is not the intention and not the case. Both methods are valid and usable, however, answering different questions.

**Reviewer Comment:**
*Other issues: (iv) The method is described as low cost and non-intrusive, but the only*
*global example given is for T42 resolution (2.8 deg). This is very low resolution for*
*current global models, yet this is only a GMD paper to establish the development of the*
*model. OK, but can you run at T159 (1.1 deg) for example with all the memory requirements*
*for the tagged tracers to be transported? I had thought that tracer transport was*
*one of the dominant costs of high-res CTMs. Indeed, line 564 seems to indicate that*
*you already have memory limitations at T42.*
**Authors' Comment:**
Actually, EMAC is not a CTM. I am not quite sure, with which coupled troposphere-to-mesosphere chemistry-climate model, multi-decadal simulations were performed at T159. The most recent models, which will participate in the upcoming IPCC via CCMI have actually a similar resolution as we have used in our study (Morgenstern et al., 2017; their table 3). Moreover, the sensitivity study presented in Section 5.1 has a horizontal resolution of 10 km or 0.1°.

**Reviewer Comment:**
*(v) What was the STE flux of O3 and NOy as a function of latitude and season. This*
*would seem to be very important since the background atmosphere shares the attribution*
*in this scheme. Please denote.*
**Authors' Comment:**
We have estimated the net ozone flux from the stratosphere in the EMAC model for the years 2000-2004. Using the residuum method, i.e. STE=Burden change-Prod+Loss+Deposition, the ozone STE is calculated to be 393 +- 25 TgO3 per year (Jöckel et al., 2006), which is in agreement with other modeling studies (Stevenson et al., 2006). Note that this method includes net fluxes, i.e. upward and downward fluxes of ozone through the tropopause. The calculation of the stratosphere-to-troposphere ozone flux was performed by using a diagnostic tracer Strat-O3, which is nudged to ozone in the stratosphere and experiences loss terms (chemistry and deposition) in the tropopshere, only. The accumulated loss terms provide a measure for the stratosphere-to-troposphere ozone flux and is 1198 +- 28 $TgO_3$ per year (see

also Jöckel et al., 2006, Table 2). The analysis of the impact of STE on the composition of the troposphere is indeed an important question, but beyond the scope of this paper. Generally, it was shown with a similar approach used here that stratospheric ozone changes and STE variations lead to variations in tropospheric ozone (e.g. Grewe, 2007). The concentration of the tagged tracer for stratospheric ozone is basically a result of the ozone produced by $O_2$ photolysis, transport to the tropopshere and tropspheric loss processes.

**Reviewer Comment:**
*(vi) The idea that the Mediterranean Sea contains "pristine areas" anywhere is at least humorous - thanks.*
**Authors' Comment:**
You're welcomed. Text adapted.

Reference:
Morgenstern, O., Hegglin, M. I., Rozanov, E., O'Connor, F. M., Abraham, N. L., Akiyoshi, H., Archibald, A. T., Bekki, S., Butchart, N., Chipperfield, M. P., Deushi, M., Dhomse, S. S., Garcia, R. R., Hardiman, S. C., Horowitz, L. W., Jckel, P., Josse, B., Kinnison, D., Lin, M., Mancini, E., Manyin, M. E., Marchand, M., Marcal, V., Michou, M., Oman, L. D., Pitari, G., Plummer, D. A., Revell, L. E., Saint-Martin, D., Schofield, R., Stenke, A., Stone, K., Sudo, K., Tanaka, T. Y., Tilmes, S., Yamashita, Y., Yoshida, K., and Zeng, G.: Review of the global models used within phase 1 of the ChemistryClimate Model Initiative (CCMI), Geosci. Model Dev., 10, 639-671, doi:10.5194/gmd-10-639-2017, 2017.